# Effectiveness of treat-to-target cholesterol-lowering interventions on cardiovascular disease and all-cause mortality risk in the community-dwelling population: a target trial emulation

Zhao Yang ⓘ, Qiujv Deng, Yongchen Hao, Na Yang, Lizhen Han, Pingping Jia ⓘ, Pan Zhou, Yiming Hao, Ziyu Wang, Wenlang Zhao, Yue Qi & Jing Liu ⓘ ✉

Little is known about the long-term effectiveness of risk-based treat-to-target cholesterol-lowering interventions on cardiovascular risk. Here, we show the emulated effectiveness of guideline-recommended low-density and non-high-density lipoprotein cholesterol-lowering interventions using the absolute risk reduction (ARR) and the restricted mean event-free time-based number needed to treat (NNT). With 5,375 participants, the 29-year risks for cardiovascular disease (CVD), all-cause mortality, and atherosclerotic CVD were 18.6%, 25.6%, and 17.7%, respectively. Long-term treat-to-target interventions showed significant reductions in CVD (ARR −2.3%, 95%CI −3.4% to −0.8%; NNT 115), all-cause mortality (−3.0%, −4.3% to −1.8%; 95), and atherosclerotic CVD (−2.6%, −3.5% to −1.2%; 104). Such effects appear more pronounced in women, smokers, and those with body mass index < 24 kg/m² or higher adherence rates.

Cardiovascular disease (CVD) dominated by atherosclerotic CVD (ASCVD) remains the leading cause of disability, mortality, and impaired quality of life in China[1], despite the advances in effective therapies targeting modifiable risk factors such as high plasma levels of low-density lipoprotein cholesterol (LDL-C)[2–4]. To date, reducing cholesterol levels has consistently lowered the risk of developing CVD in a log-linear dose-response fashion[5–7]. For instance, per 1 mmol/L (i.e., 39 mg/dL) LDL-C reduction reduces the relative major vascular risk by 21%, and such an effect is more evident for individuals at high cardiovascular risk[7]. Thus, current guidelines for primary prevention recommend risk-based cholesterol-lowering interventions to reduce LDL-C levels[2,8,9] or nonhigh-density lipoprotein cholesterol (non-HDL-C)[4,10] to prevent or delay the occurrence of CVD[11,12].

However, little is known about the long-term effectiveness of risk-based cholesterol-lowering interventions on CVD and all-cause mortality, given the time-evolving risk profiles and risk stratification after the intervention[13,14], referred to as treat-to-target effects[15]. This is particularly true for risk-based cholesterol-lowering interventions at an isolated time point[16]. Recently, time course or cumulative exposure of cholesterol profiles[17–19] and Mendelian randomization studies[20–23] have been proposed to assess the long-term or even lifetime effects of cholesterol-lowering interventions on cardiovascular risk. However, these studies used either standard regression, conditioning methods, or genetic instruments with strong assumptions for confounding control, inheriting methodological limitations and limiting their capability to quantify the effect of time-evolving interventions[15,24–27]. For example, cardiovascular risk derived from traditional risk factors[2] at a specific time point often guides future cholesterol-lowering interventions. However, cardiovascular risk itself is also affected by past interventions and risk profiles. Then, the cholesterol-lowering

---

Center for Clinical and Epidemiologic Research, Beijing Anzhen Hospital, Capital Medical University, Beijing Institute of Heart, Lung and Blood Vessel Diseases, Beijing, People's Republic of China. ✉e-mail: jingliu@ccmu.edu.cn

interventions could subsequently influence cardiovascular risk profiles. Mendelian randomization estimates usually provide qualitative information for the effect direction of a clinical intervention that might not correspond to the magnitude of the effect in practice[28]. More importantly, most of these studies over-emphasized relative risk rather than absolute risk, which is more appropriate in primary care from a public health perspective. Lastly, treat-to-target cholesterol-lowering interventions involve previous risk profiles and intervention effects at each follow-up visit, conferring per-protocol effects. Thus, the estimation of treat-to-target effects requires accounting for time-varying confounding feedback. Sequentially randomized controlled experiments randomly assigned treat-to-target cholesterol-lowering interventions at each visit during follow-up according to previous risk profiles and intervention effects can appropriately address time-varying confounding by adjusting for a set of minimal sufficient confounders, but this protocol is typically not feasible for practical and ethical reasons[29].

The use of longitudinal data to emulate a randomized (hypothetical) pragmatic target trial[30,31], alongside parametric g-formula, offers a practical approach to adjusting time-varying confounding[25,26] and inferring causal effects of dynamic joint cholesterol-lowering interventions on CVD and all-cause mortality under several untestable and often plausible assumptions. Thus, the rationale of this study is to emulate a target trial to answer the following research question: What were the long-term treat-to-target effects on the primary prevention of CVD and all-cause mortality in the community-dwelling population if all people continually adhered to the risk-based cholesterol-lowering targets recommended by clinical guidelines[2,32], based on the Chinese Multi-provincial Cohort Study (CMCS), a community-dwelling prospective cohort.

## Results
We explicitly emulated the target trial of long-term treat-to-target cholesterol-lowering interventions and cardiovascular events (Table 1) using the 29-year follow-up data from CMCS. We mimicked each protocol component, with necessary modifications to appropriate handling time-varying intervention and confounding via the parametric g-formula (see "Methods").

### Study participants and characteristics
Figure 1 presents the flowchart of participants, with details on CMCS participants described in the "Method" section. Overall, a total of 5735 eligible CMCS participants aged ≥ 35 years, with no prior CVD history and no cholesterol levels missing at baseline during 1992-1993 were included. Table 2 shows the characteristics of eligible CMCS participants. The mean age at baseline was 47 years, and 61% were women. The mean LDL-C and non-HDL-C at baseline were 2.7 mmol/L and 3.3 mmol/L, respectively, with 1.4% taking cholesterol-lowering medications and 8.9% at high risk of developing ASCVD. Over the study period from 1992 to 2020, the mean LDL-C and non-HDL-C increased, along with a remarkably increased proportion of taking cholesterol-lowering medications and a higher proportion of participants at intermediate-to-high cardiovascular risk.

Until the end of 2020 (157,257.8 person-years), 1203 CVD, 1621 all-cause mortality, and 1171 ASCVD were recorded. Figure 2 shows the observed 29-year cumulative risk, compared to the simulated risk under the natural course of no intervention for CVD, all-cause mortality, and ASCVD. Notably, the parametric g-formula estimated risk over the 29-year follow-up period was close to the observed risk, with 20.9% vs. 20.1% for CVD, 28.3% vs. 26.5% for all-cause mortality, and 20.0% vs. 19.2% for ASCVD, respectively.

### Estimated effectiveness of treat-to-target interventions
We estimated the effectiveness of treat-to-target cholesterol-lowering interventions via a parametric g-formula estimator (Supplementary

Table 1 and Fig. 3). Supplementary Table 2 shows the estimated 29-year cumulative risk, cumulative risk difference, and the restricted mean of event-free time (RMET)-based number needed to treat (NNT) to prevent one CVD, all-cause mortality, and ASCVD event, comparing the natural course of no intervention, with Supplementary Table 3 showing the relative risk reduction regarding risk ratio. The estimated observational analogy of per-protocol effect was associated with a relative risk reduction of 12% for CVD (risk ratio: 0.88; 95% CI: 0.82 to 0.96), 12% for all-cause mortality (0.88; 0.84 to 0.93), and 15% for ASCVD (0.85; 0.81 to 0.93), which corresponded to an absolute risk of −2.3% (risk difference: 95% CI: −3.4% to −0.8%), −3.0% (−4.3% to −1.8%), −2.6% (−3.5% to −1.2%) and NNT of 115, 95, and 104 (Fig. 4), respectively. Furthermore, treat-to-target interventions on subgroups of women (ARR −2.5% vs. −0.8%; NNT 112 vs. 336), BMI < 24 kg/m² (−2.9% vs. −1.0%; 91 vs 252), and smokers (−3.5% vs. −1.0%; 75 vs. 271) yielded more evident benefit for CVD, of women (−2.9% vs. −1.6%; 102 vs. 166), BMI < 24 kg/m² (−4.6% vs. −0.7%; 62 vs. 367) and smokers (−3.0% vs. −2.0%; 88 vs. 139) for all-cause mortality, and of women (−2.6% vs. −2.0%; 99 vs. 247) and smokers (−3.7% vs. −1.3%; 73 vs. 211) for ASCVD, as detailed in Fig. 4. Of these benefits, roughly one-fifth of CMCS participants underwent treat-to-target interventions over the study period, as presented in Supplementary Table 3.

### Estimated effectiveness of feasible interventions
Consistent results were also observed under feasible interventions defined as 80% of eligible participants receiving the corresponding intervention when the risk-based conditions were met over the study period, compared to the natural course of no interventions, as detailed in Supplementary Table 4 and Fig. 5. Keeping an adherence rate of 80% over the study period yielded a relative risk reduction of 12% (risk ratio: 0.88; 95% CI: 0.84 to 0.96) for CVD, 11% (0.89, 0.84 to 0.94) for all-cause mortality, and 14% (0.86, 0.82 to 0.94) for ASCVD, corresponding to an absolute risk reduction of −2.1% (risk reduction: 95% CI: −3.1% to −0.7%), −2.8% (95% CI: −4.1% to −1.7%), −2.4 (95% CI: −3.3 to −1.1) and NNT of 124, 100, and 112, with ~21% of CMCS participants underwent the feasible interventions (Supplementary Table 5). Similarly, more evident benefits were noted among subgroups of women, BMI < 24 kg/m², and smokers (Fig. 5).

### Sensitivity analyses and positive/negative control analyses
When allowing the time-delayed effects (i.e., reordering time-varying variables) of treat-to-target cholesterol-lowering interventions on CVD, all-cause mortality, and ASCVD, the estimated parametric g-formula risks decreased slightly, toward the null (Fig. 4 and Fig. 5). When decreasing adherence rate from 70% to 20%, the protective effects of treat-to-target intervention deteriorated for CVD, all-cause mortality, and ASCVD, particularly for the NNT. When specifying and emulating a target trial of statin therapy and major vascular disease using the CMCS data, a per-protocol risk ratio of 0.83 (95% CI: 0.68 to 0.95) for CVD, 0.83 (95% CI: 0.69 to 0.91) for all-cause mortality, and 0.78 (95% CI: 0.63 to 0.90) for ASCVD were observed by comparing static LDL-C-lowering by 1 mmol/L versus no interventions over the study period, as detailed in Supplementary Table 3. Similarly, no cholesterol-lowering effects on cancer mortality were noted (Table 3).

## Discussion
This study emulates a target trial via a 29-year community-dwelling-based longitudinal cohort of 5735 CMCS participants to assess the effectiveness of long-term treat-to-target cholesterol-lowering interventions adhering to the 2020 CSC recommendations or feasible interventions on CVD, all-cause mortality, and ASCVD. We found that adhering to risk-based treat-to-target cholesterol-lowering intervention for 29 years reduced 2.3% absolute risk of CVD, 3.0% of all-cause mortality, and 2.6% of ASCVD in the community-dwelling population, respectively, compared to no interventions, equivalent to REMT-based

**Table 1 | Specification and emulation of a target trial of treat-to-target cholesterol-lowering interventions to prevent cardio-vascular disease and all-cause mortality**

| Protocol component | Target trial specification | Target trial emulation |
|---|---|---|
| **Eligibility criteria** | Aged 35 years or older between January 1, 1992 and December 31, 1993<br>In-person examination at the baseline<br>No history of cardiovascular disease at the baseline<br>Triglyceride < 4.52 mmol/L (400 mg/dL) at baseline, defined as the recruitment date of study participants | Same as for the target trial<br>We defined a history of cardiovascular disease as a composite outcome comprising heart disease and stroke before the date of recruitment in the Chinese Multi-provincial Cohort Study |
| **Treatment strategies** | The natural course of no intervention<br>Treat-to-target cholesterol-lowering intervention[a]<br>Feasible treat-to-target cholesterol-lowering intervention, defined as 80% of eligible participants receiving the intervention at the follow-up examination over the study period | Same as for the target trial |
| **Treatment assignment** | Participants are randomly assigned to an intervention strategy at baseline or during the follow-up when the risk-based conditions are met | We implemented the hypothetical treat-to-target intervention by transforming the follow-up time into a one-year unit and initiating the hypothetical interventions when the risk-based conditions were met |
| **Outcomes** | The primary outcomes are cardiovascular disease and all-cause mortality<br>The secondary outcome is atherosclerotic cardiovascular disease<br>Competing events are defined as non-cardiovascular deaths before the occurrence of the outcome of interest (except for all-cause mortality) | Same as for the target trial |
| **Follow-up** | The follow-up period starts at baseline and ends at the year of recording cardiovascular deaths, non-cardiovascular deaths, loss to follow-up, 29 years after baseline, or administrative end of follow-up on 31 December 2020, whichever occurs first | Same as for the target trial<br>We defined the start of the follow-up period (i.e., time zero) as the initiation time of the intervention when the risk-based conditions are met |
| **Causal contrasts** | Per-protocol effect | Observational analog of per-protocol effect |
| **Statistical analysis** | Per-protocol analysis of the cholesterol-lowering effects under various strategies<br>Competing risk analysis of the total cholesterol-lowering effects in the presence of competing events<br>Subgroup analyses by sex (men *vs.* women), BMI ( < 24 *vs.* ≥24 kg/m$^2$), smoking status (Yes *vs.* No), and anti-hypertensive drugs (Yes *vs.* No) at the baseline<br>Sensitivity analysis of the cholesterol-lowering effects under various model specifications and a range of adherence rates from 70% to 20%<br>Positive and negative control analyses of replicating the constant ~21% cardiovascular risk reduction with statin therapy per 1 mmol/L (39 mg/dL) reduction in plasma LDL-C levels and the null association of cholesterol-lowering interventions with cancer deaths. | Same per-protocol analysis with sequential emulation and adjustment for baseline covariates<br>Same competing risk analyses<br>Same subgroup analyses<br>Same sensitivity and positive/negative control analyses |

Note: [a]Treat-to-target cholesterol-lowering intervention is based on cholesterol-lowering targets recommended by the Chinese Society of Cardiology in 2020 on LDL-C and non-HDL-C levels, i.e., for participants with diabetes at high cardiovascular risk, lower the LDL-C to < 1.8 mmol/L (70 mg/dL, i.e., a fixed level drawn from a uniform distribution with a upper bound of 1.8 mmol/L) or LDL-C reduction to > 50% from baseline whichever is the lowest and non-HDL-C to < 2.6 mmol/L (100 mg/dL, i.e., a fixed level drawn from a uniform distribution with a upper bound of 2.6 mmol/L); for participants without diabetes who are at moderate-to-high cardiovascular risk lower the LDL-C to < 2.6 mmol/L (100 mg/dL, i.e., a fixed level drawn from a uniform distribution with a upper bound of 2.6 mmol/L) and non-HDL-C to < 3.4 mmol/L (130 mg/dL, i.e., a fixed level drawn from a uniform distribution with a upper bound of 3.4 mmol/L); for participants at low cardiovascular risk, lower LDL-C to < 3.4 mmol/L (130 mg/dL, i.e., a fixed level drawn from a uniform distribution with a upper bound of 3.4 mmol/L) and a non-HDL-C < 4.2 mmol/L (160 mg/dL, i.e., a fixed level drawn from a uniform distribution with a upper bound of 4.2 mmol/L).

NNT of 115, 95, 104 to prevent one CVD, all-cause mortality, and ASCVD event. Furthermore, the higher adherence rate of risk-based treat-to-target cholesterol-lowering intervention would confer more evident benefit.

Given that no randomized controlled trials compared the preventive effects of time-evolving risk-based cholesterol-lowering interventions with no interventions, we cannot directly verify the preventive effects from the hypothetical treat-to-target intervention. However, our hypotheses derived from the current guidelines on the primary prevention of cardiovascular disease[2,8,9], synthetic evidence from statin therapy randomized trials showing a clear net benefit in people at low risk of vascular disease[7,33], cardiovascular-risk-based approaches for determining statin eligibility in primary prevention[34,35], and cumulative cholesterol profiles exposure showing an increased lifetime cardiovascular risk[17–19]. More importantly, our positive control analysis emulating a target trial of statin therapy and vascular disease yielded a constant of 22% ASCVD relative risk reduction per 1 mmol/L reduction in plasma LDL-C levels *versus* no intervention, in line with the Cholesterol Treatment Trialists meta-analysis of statin therapy in people at low vascular risk from 27 randomized trials showing a 21% relative risk reduction in major cardiovascular event[7].

Similarly, our findings were consistent with the pooled analysis from the US Preventive Services Task Force, showing a 1.3% absolute risk reduction after a mean follow-up of 3.3 years (ranging from 6 months to 6 years of follow-up)[33], enhancing the credibility of our results. However, when translating such protective effects into NNT in public health, our findings conferred more benefit (RMET-based NNT 65 for ASCVD) per 1 mmol/L LDL-C reduction than the Cholesterol Treatment Trialists meta-analysis of statin therapy (absolute risk reduction [ARR]-based NNT 130)[7] to prevent one major cardiovascular event. Such a discrepancy may lie in that the ARR-based NNT failed to capture the profile of therapeutic effect as it is calculated at a specific time point while the RMET-based NNT quantified the therapeutic effect via average event-free time, providing a more accurate, intuitive, and clinically meaningful interpretation[36]. Nevertheless, our findings were consistent with the pooled analysis from the US Preventive Services Task Force showing an ARR-based NNT of 77[31] and the individualized statin benefits via absolute risk reduction estimated from

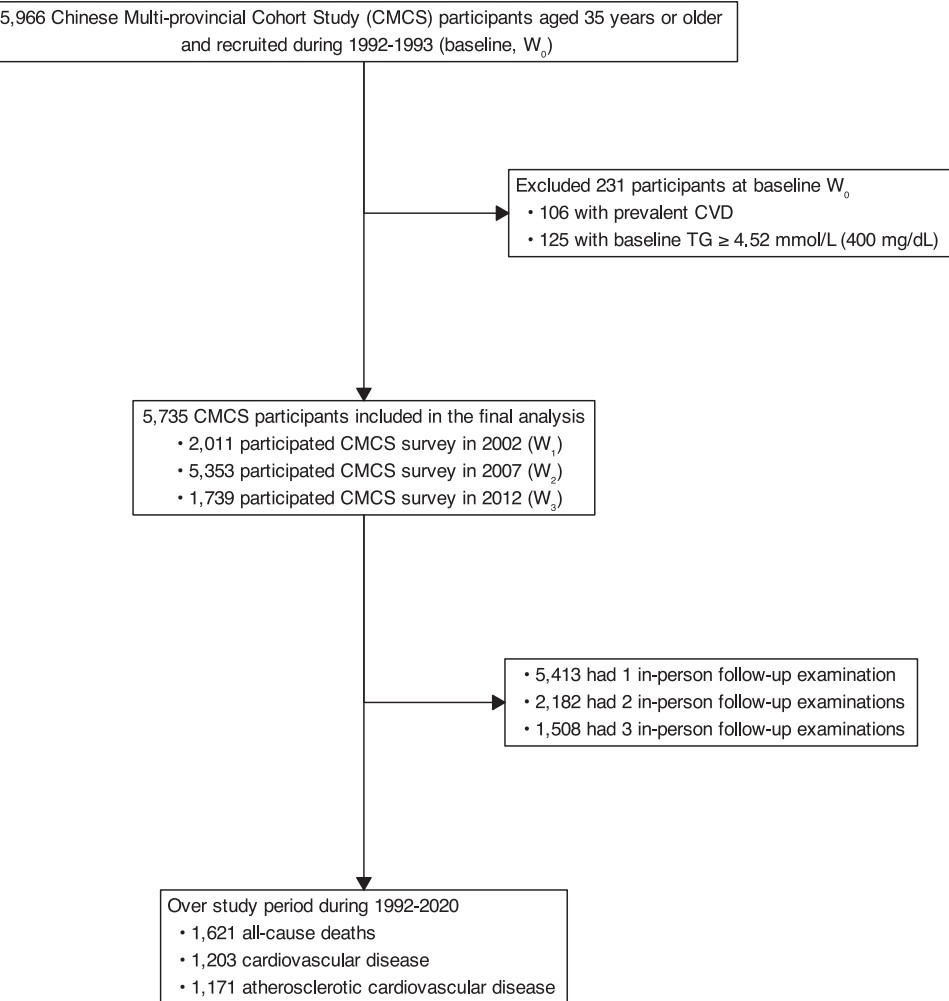

**Fig. 1 | Flowchart of study participants' selection.** Flowchart of selecting study participants from the Chinese Multi-provincial Cohort Study 1992–2020.

population-based risk prediction models with a range of 10-year predicted ARR-based NNT from 44 to 67[34,35,37,38].

Furthermore, previously prospective cohort studies using the traditional Cox model showed LDL-C cumulative exposure in early life yielded a 4.1% to 5.3% higher relative risk of developing CVD in late life[17,18], which was smaller than our results. Such a discrepancy could be attributed to the additional adjustment for LDL-C at the end of the observation window, blocking the effect of cumulative LDL-C exposure on the outcome and inducing a downward bias toward the null[19,25]. Conversely, our study explicitly specifies the protocol of the target trial and emulates it using the longitudinal data in the CMCS cohort with a precise, treat-to-target intervention specification and appropriate protective effect estimation via parametric g-formula, essentially addressing these biases.

Lastly, our subgroup analyses showed mild absolute risk reduction among participants with antihypertensive treatment, possibly resulting from immortal time or survival bias[31] since those eligible hypertension participants with treat-to-target interventions had to survive without cardiovascular disease and remain under follow-up for a long time, as noted and explained in our previous study[39].

To the best of our knowledge, this is the first pragmatic target trial emulating the hypothetical time-evolving cholesterol-lowering interventions and showing that long-term risk-based treat-to-target cholesterol-lowering interventions could exert an equivalent protective effect concerning the statin therapeutic trials on cardiovascular risk in the community-dwelling population, particularly when

maintaining the adherence rate at a high level (e.g., 70%–80%). Moreover, our study has several additional strengths. The long-term follow-up with multiple waves of in-person follow-up examinations and a large number of covariates in the prospective CMCS cohort enables us to quantify the absolute and long-term effectiveness of adhering treat-to-target cholesterol-lowering interventions on CVD after adjusting for time-evolving cardiovascular risk profiles and interventions over the study period that may not be easily conducted in a randomized trial. We also considered competing events in our analyses to mimic how such effects will likely be affected by the treat-to-target cholesterol-lowering interventions via deaths from other causes.

However, our study also had some limitations. First, a central challenge of specifying and emulating a target trial for target-to-target interventions via parametric g-formula is the strong assumption of no unmeasured confounding depending on the past intervention and covariate history over the study period, which is often not guaranteed to hold in an observational study[40]. For example, in current analyses, we couldn't adjust for some other confounders, such as the history of major chronic diseases (e.g., chronic kidney disease [CKD], chronic liver disease, and chronic obstructive pulmonary disease), alcohol use, physical activity, and dietary habit, mainly due to their impact, except for CKD, on CVD often through the classical cardiovascular risk factors[41]. Though CKD has been increasingly recognized as an important risk factor for CVD in recent years, it was not considered in the Chinese guidelines till 2020[2,42]. Moreover, data on CKD and liver

**Table 2 | Characteristics of study participants from the Chinese Multi-provincial Cohort Study stratified by waves of follow-up examination. Values are numbers (percentages) unless stated otherwise**

| Characteristics | Wave 0 (year 1992) | Wave 1 (year 2002) | Wave 2 (year 2007) | Wave 3 (year 2012) |
|---|---|---|---|---|
| Total no. of participants | 5735 | 2011 | 5353 | 1739 |
| Age at entry, years (SD) | 46.6 (7.7) | 46.8 (8.0) | 46.3 (7.6) | 46.2 (7.9) |
| Age at examination, years (SD) | 46.9 (7.7) | 57.2 (7.9) | 61.8 (7.5) | 67.0 (7.8) |
| Sex women | 3477 (61%) | 1181 (59%) | 2797 (52%) | 1042 (60%) |
| Attained education | | | | |
| College or above | 901 (16%) | 368 (35%) | 1086 (26%) | 333 (36%) |
| Junior college or less | 4834 (84%) | 697 (65%) | 3086 (74%) | 590 (64%) |
| Occupation | | | | |
| Manual labor | 3905 (68%) | 1041 (52%) | 2415 (45%) | 908 (52%) |
| Non-manual labor | 1830 (32%) | 970 (48%) | 2938 (55%) | 831 (48%) |
| Current smoker | 1389 (24%) | 240 (12%) | 959 (18%) | 145 (8.3%) |
| Body mass index, kg/m² (SD) | 23.9 (3.1) | 25.4 (3.3) | 24.8 (3.4) | 24.7 (3.3) |
| Systolic blood pressure, mm Hg (SD) | 122.2 (18.8) | 130.8 (18.8) | 139.0 (20.5) | 136.8 (16.3) |
| Diastolic blood pressure, mm Hg (SD) | 79.5 (11.7) | 81.9 (10.3) | 83.3 (10.8) | 79.2 (9.3) |
| Total cholesterol, mmol/L (SD) | 4.6 (0.9) | 5.4 (1.0) | 5.2 (1.0) | 5.3 (1.0) |
| High-density lipoprotein cholesterol, mmol/L (SD) | 1.4 (0.3) | 1.4 (0.3) | 1.4 (0.3) | 1.3 (0.3) |
| Low-density lipoprotein cholesterol, mmol/L (SD) | 2.7 (0.8) | 3.3 (0.8) | 3.3 (0.9) | 3.1 (0.8) |
| Triglyceridemic, mmol/L (Median, IQR) | 1.2 (0.8, 1.6) | 1.3 (0.9, 1.9) | 1.4 (1.1, 2.0) | 1.4 (1.0, 1.9) |
| Non-HDL-C, mmol/L (SD) | 3.3 (0.9) | 4.0 (1.0) | 3.8 (0.9) | 4.0 (1.0) |
| Fasting blood glucose, mmol/L (SD) | 3.6 (1.7) | 5.0 (1.2) | 5.5 (1.4) | 5.6 (1.2) |
| Diabetes | 114 (2.0%) | 211 (10%) | 763 (14%) | 308 (18%) |
| Antihypertensive drugs | 286 (5.0%) | 463 (23%) | 1576 (29%) | 1095 (63%) |
| Cholesterol-lowering drugs | 78 (1.4%) | 137 (6.8%) | 320 (6.0%) | 784 (45%) |
| Cardiovascular risk stratification[a] | | | | |
| Low | 4665 (81.3%) | 1084 (54%) | 2266 (42%) | 422 (24%) |
| Intermediate | 569 (9.9%) | 478 (24%) | 1648 (31%) | 818 (47%) |
| High | 511 (8.9%) | 449 (22%) | 1439 (27%) | 499 (29%) |

Note: [a] The cardiovascular risk stratification was assessed following the ASCVD and CVD risk algorithms recommended by the Chinese guideline for the primary prevention of cardiovascular disease, in which the 10-year risk stratification of total cardiovascular risk is the same as the atherosclerotic cardiovascular disease in most cases. Beirfly, a 3-step evaluation procedure was employed: (1) participants with diabetes (aged ≥40 years) or LDL-C ≥4.9 mmol/L (or total cholesterol [TC] ≥7.2 mmol/L) were directly classified at high risk; (2) participants who do not meet procedure (1), sex-spexific 10-year ASCVD risk assessment algorithms, including LDL-C or TC levels, hypertension, smoking status, HDL-C, and age ≥45/55 years (male/female), were used, based on which 10-year ASCVD risk <5%, 5% – 9%, and ≥10% were defined as low, intermediate, and high risk, respectively; (3) for those intermediate-risk participants aged <55 years, the lifetime risk of cardiovascular disease were assessed, in which participants with ≥2 following risk factors are defined at high risk: a) systolic blood pressure ≥160 mmHg or diastolic blood pressure ≥100 mmHg; b) non-HDL-C ≥5.2 mmol/L (200 mg/dL); c) HDL-C <1.0 mmol/L (40 mg/dL); d) body mass index (BMI) ≥28 kg/m²; and e) smoking.

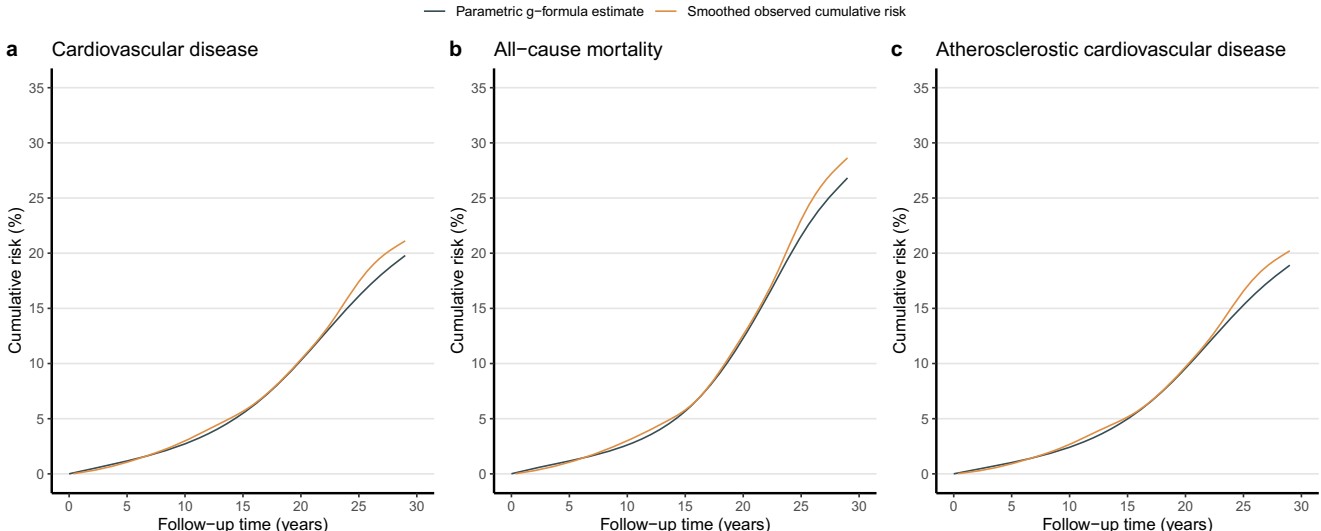

**Fig. 2 | Smoothed observed cumulative risk *versus* simulated risk under no intervention.** The smoothed observed cumulative risk (orange) compared to simulated risk (dark blue) under the natural course of no intervention for cardiovascular disease (**a**), all-cause mortality (**b**), and atherosclerotic cardiovascular disease (**c**).

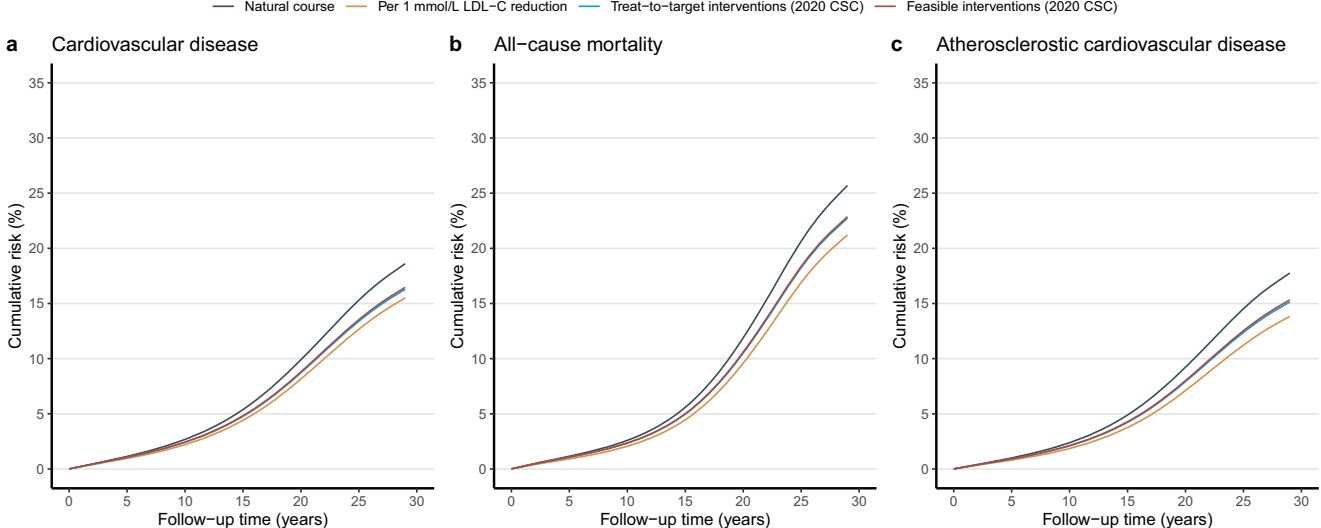

**Fig. 3 | Estimated cumulative risk curves under hypothetical interventions.** Estimated cumulative risk curves comparing treat-to-target hypothetical interventions (i.e., per 1 mmol/L LDL-C reduction (orange), treat-to-target (cyan), and feasible interventions (dark red)) with the "natural course" of no interventions (dark blue) for cardiovascular disease (**a**), all-cause mortality (**b**), and atherosclerotic cardiovascular disease (**c**) based on the Chinese Multi-provincial Cohort Study 1992-2020. Note: [†] Treat-to-target cholesterol-lowering intervention is based on cholesterol-lowering targets recommended by the Chinese Society of Cardiology in 2020 on LDL-C and non-HDL-C levels, i.e., for participants with diabetes at high cardiovascular risk, lower the LDL-C to < 1.8 mmol/L (70 mg/dL, i.e., a fixed level drawn from a uniform distribution with a upper bound of 1.8 mmol/L) or LDL-C reduction to > 50% from baseline whichever is the lowest and non-HDL-C to < 2.6 mmol/L (100 mg/dL, i.e., a fixed level drawed from a uniform distribution with a upper bound of 2.6 mmol/L); for participants without diabetes who are at moderate-to-high cardiovascular risk lower the LDL-C to < 2.6 mmol/L (100 mg/dL, i.e., a fixed level drawed from a uniform distribution with a upper bound of 2.6 mmol/L) and non-HDL-C to < 3.4 mmol/L (130 mg/dL, i.e., a fixed level drawed from a uniform distribution with a upper bound of 3.4 mmol/L); for participants at low cardiovascular risk, lower LDL-C to < 3.4 mmol/L (130 mg/dL, i.e., a fixed level drawed from a uniform distribution with a upper bound of 3.4 mmol/L) and a non-HDL-C < 4.2 mmol/L (160 mg/dL, i.e., a fixed level drawed from a uniform distribution with a upper bound of 4.2 mmol/L). [‡] Feasible treat-to-target cholesterol-lowering intervention, defined as 80% of eligible participants receiving the intervention at the follow-up examination over the study period.

diseases was unavailable at the baseline. However, the prevalence of CKD 3–4 (defined as eGFR< 60 ml·min$^{-1}$·(1.73 m$^2$)$^{-1}$) and elevated liver function (defined as alanine aminotransferase and/or aspartate aminotransferase is elevated ≥ 3 times the upper limit of normal and combined with elevated total bilirubin) were only 3.2% and 0.3% in 2002 visit in CMCS; thus their impact on our results would be minimal.

Second, the parametric g-formula relies on the correct model specification. The divergence between the observed cumulative cardiovascular risk and the parametric g-formula estimated risk under the natural course after a 25-year follow-up suggests potential model misspecifications (Fig. 2). However, the consistent effects using the doubly-robust estimators triangulate our results, which were less likely to be affected by model misspecification (Table 4)[43,44]. Third, the interpretation of our causal estimates also relies on the well-defined interventions and consistency assumption—the counterfactual outcome under the observed intervention value is equal to the observed outcome for each subject. However, we didn't explain how cholesterol levels would be lowered, which means the observed protective estimates are overall interventional effects, making the consistency assumption dubious in the current study[45]. Therefore, our estimates should be interpreted as an overall effect of various LDL-C lowering strategies in the CMCS population[46]. Further studies with detailed assessments of cholesterol-lowering interventions remain needed. Fourth, we did not assess the possible protective effects of treat-to-target interventions on each component of CVD owing to the small number of each event that could be competing events for other outcomes of interest. Fifth, our results should be interpreted carefully concerning the possible side effects of cholesterol-lowering interventions, particularly aggressive cholesterol-lowering manner[47], even if our previous study had shown such side effects likely resulted from survival bias[39]. Finally, our findings might have been underpowered

and imprecise due to the relatively small sample size in wave 1 and wave 3[48] and large time interval over the study period. However, we included all eligible participants from multiple provinces in wave 0 and wave 4 from the CMCS cohort to avoid selection bias and achieve larger statistical power alongside more precise treat-to-target estimates. A large-scale longitudinal study with more frequent measurements of traditional cardiovascular risk profiles, lifestyle, medication, and disease history (e.g., major chronic diseases) would further triangulate our results.

Nevertheless, our findings have important public health implications since contemporary guides recommend risk-based cholesterol-lowering in primary and secondary prevention of CVD at an isolated time point based on the updated evidence and methods[8,9]. Nowadays, lifetime interventions for cholesterol-lowering have also been advocated for CVD prevention[16], mainly from Mendelian randomization studies[20,21]. From this point, our study provides first-hand evidence showing that long-term treat-to-target cholesterol-lowering interventions could reduce CVD, all-cause mortality, and ASCVD risk in the community population. For instance, 115 participants receiving the treat-to-target interventions would prevent one CVD event, with an average of 27.2-year event-free survival time to 29 years of follow-up, particularly in women, BMI < 24 kg/m², and smokers, with NNTs varying from 75 to 112 participants. Notably, such protective effects were more evident when increased the adherence rate of treat-to-target cholesterol-lowering interventions from 20% to 70%, yielding roughly a halved NNT from 279 to 131 participants, suggesting the necessity of a high adherence rate to maintain the maximum benefit from treat-to-target cholesterol-lowering interventions.

Here, we show that long-term treat-to-target cholesterol-lowering interventions in the community population could lower the 29-year risk of developing CVD, all-cause mortality, and ASCVD, compared to

| Intervention | CVD Risk (%, 95%CI) | CVD RD (%, 95%CI) | NNT | All-cause mortality Risk (%, 95%CI) | All-cause mortality RD (%, 95%CI) | NNT | ASCVD Risk (%, 95%CI) | ASCVD RD (%, 95%CI) | NNT |
|---|---|---|---|---|---|---|---|---|---|
| **Natural course** | 18.6 (17.7 to 19.9) | | | 25.6 (24.7 to 27.4) | | | 17.7 (16.9 to 19.0) | | |
| **Treat-to-target interventions** | 16.3 (15.4 to 18.3) | -2.3 (-3.4 to -0.8) | 115 | 22.7 (21.8 to 25.2) | -3.0 (-4.3 to -1.8) | 95 | 15.1 (14.2 to 17.2) | -2.6 (-3.5 to -1.2) | 104 |
| **Subgroup analyses** | | | | | | | | | |
| **Sex** | | | | | | | | | |
| Women | 14.4 (11.2 to 17.1) | -2.5 (-5.2 to 0.0) | 112 | 19.5 (15.2 to 22.9) | -2.9 (-6.3 to 0.2) | 102 | 13.2 (10.2 to 15.7) | -2.9 (-5.4 to -0.0) | 99 |
| Men | 19.2 (16.6 to 22.6) | -0.8 (-3.3 to 2.3) | 336 | 28.9 (25.5 to 32.6) | -1.6 (-4.4 to 1.2) | 166 | 18.1 (15.5 to 21.4) | -1.0 (-3.6 to 1.8) | 247 |
| **BMI (kg/m²)** | | | | | | | | | |
| < 24 | 12.1 (10.0 to 15.2) | -2.9 (-5.2 to 0.5) | 91 | 17.6 (14.2 to 21.2) | -4.6 (-7.8 to -1.0) | 62 | 11.6 (9.7 to 14.8) | -2.6 (-4.9 to 1.0) | 99 |
| ≥ 24 | 22.5 (19.4 to 25.0) | -1.0 (-4.4 to 1.5) | 252 | 30.0 (25.9 to 34.1) | -0.7 (-3.5 to 3.5) | 367 | 20.5 (17.7 to 22.9) | -2.0 (-5.1 to 0.2) | 134 |
| **Smoking status** | | | | | | | | | |
| Yes | 20.8 (17.6 to 27.3) | -3.5 (-6.0 to 1.0) | 75 | 32.5 (26.1 to 41.6) | -3.0 (-8.2 to 2.7) | 88 | 19.9 (16.7 to 25.6) | -3.7 (-5.9 to 0.4) | 73 |
| No | 16.3 (13.9 to 19.3) | -1.0 (-2.8 to 1.6) | 271 | 21.4 (18.6 to 24.2) | -2.0 (-4.5 to 0.6) | 139 | 15.2 (12.9 to 18.3) | -1.3 (-3.2 to 1.4) | 211 |
| **Antihypertensive medication** | | | | | | | | | |
| Yes | 35.7 (26.2 to 42.0) | 2.8 (-2.5 to 10.0) | -58 | 48.3 (36.5 to 56.8) | 6.1 (1.6 to 12.8) | -31 | 34.1 (24.8 to 39.7) | 2.0 (-3.5 to 9.0) | -76 |
| No | 16.1 (14.0 to 19.0) | -2.1 (-3.5 to 0.6) | 129 | 21.5 (19.0 to 24.4) | -3.7 (-5.7 to -1.4) | 77 | 14.9 (13.2 to 17.5) | -2.4 (-3.8 to 0.1) | 116 |
| **Sensitivity analyses** | | | | | | | | | |
| Reordering time-varying variables | 16.3 (15.4 to 18.3) | -2.3 (-3.4 to -0.8) | 116 | 22.7 (21.8 to 25.2) | -3.0 (-4.3 to -1.8) | 93 | 15.1 (14.2 to 17.1) | -2.6 (-3.5 to -1.2) | 104 |
| Adherence rate of 70% | 16.5 (15.7 to 18.6) | -2.0 (-3.0 to -0.6) | 131 | 22.9 (22.2 to 25.4) | -2.7 (-4.0 to -1.6) | 104 | 15.4 (14.5 to 17.4) | -2.3 (-3.1 to -1.0) | 118 |
| Adherence rate of 60% | 16.7 (15.9 to 18.6) | -1.9 (-2.8 to -0.5) | 142 | 23.0 (22.3 to 25.5) | -2.6 (-3.9 to -1.5) | 110 | 15.6 (14.7 to 17.5) | -2.2 (-2.9 to -0.8) | 127 |
| Adherence rate of 50% | 16.8 (16.0 to 18.6) | -1.7 (-2.6 to -0.4) | 155 | 23.2 (22.5 to 25.5) | -2.4 (-3.7 to -1.4) | 117 | 15.8 (14.8 to 17.6) | -2.0 (-2.7 to -0.7) | 139 |
| Adherence rate of 40% | 17.0 (16.2 to 18.9) | -1.5 (-2.3 to -0.3) | 175 | 23.4 (22.6 to 25.7) | -2.3 (-3.5 to -1.3) | 128 | 16.0 (15.0 to 17.8) | -1.8 (-2.5 to -0.6) | 157 |
| Adherence rate of 30% | 17.3 (16.4 to 19.1) | -1.3 (-2.0 to -0.1) | 210 | 23.6 (22.8 to 25.8) | -2.0 (-3.1 to -1.2) | 145 | 16.2 (15.3 to 18.0) | -1.5 (-2.2 to -0.4) | 188 |
| Adherence rate of 20% | 17.6 (16.7 to 19.3) | -1.0 (-1.7 to 0.0) | 279 | 24.0 (23.1 to 26.0) | -1.7 (-2.7 to -0.9) | 179 | 16.6 (15.7 to 18.2) | -1.1 (-1.8 to -0.2) | 250 |
| **Positive control analyses** | | | | | | | | | |
| Per 1 mmol/L LDL-C reduction | 15.5 (12.9 to 18.2) | -3.1 (-6.1 to -0.9) | 79 | 21.2 (17.9 to 24.8) | -4.5 (-8.2 to -2.3) | 58 | 13.8 (11.4 to 16.5) | -3.9 (-6.7 to -1.8) | 65 |

**Fig. 4 | Forest plots of the estimated effects under target-to-treat interventions.** Forest plot of the estimated parametric g-formula risk (%), risk difference (RD, %), and restricted mean event-free based number needed to treat (NNT) to prevent one cardiovascular, all-cause mortality, and atherosclerotic cardiovascular event under natural course and treat-to-target interventions recommended by the 2020 Chinese Society of Cardiology after 29 years of follow-up from the Chinese Multi-provincial Cohort Study 1992-2020. Note: [†] Treat-to-target cholesterol-lowering intervention is based on cholesterol-lowering targets recommended by the Chinese Society of Cardiology in 2020 on LDL-C and non-HDL-C levels, i.e., for participants with diabetes at high cardiovascular risk, lower the LDL-C to < 1.8 mmol/L (70 mg/dL, i.e., a fixed level drawn from a uniform distribution with a upper bound of 1.8 mmol/L) or LDL-C reduction to > 50% from baseline whichever is the lowest and non-HDL-C to < 2.6 mmol/L (100 mg/dL, i.e., a fixed level drawn from a uniform distribution with a upper bound of 2.6 mmol/L); for participants without diabetes who are at moderate-to-high cardiovascular risk lower the LDL-C to < 2.6 mmol/L (100 mg/dL, i.e., a fixed level drawn from a uniform distribution with a upper bound of 2.6 mmol/L) and non-HDL-C to < 3.4 mmol/L (130 mg/dL, i.e., a fixed level drawn from a uniform distribution with a upper bound of 3.4 mmol/L); for participants at low cardiovascular risk, lower LDL-C to < 3.4 mmol/L (130 mg/dL, i.e., a fixed level drawn from a uniform distribution with a upper bound of 3.4 mmol/L) and a non-HDL-C < 4.2 mmol/L (160 mg/dL, i.e., a fixed level drawn from a uniform distribution with a upper bound of 4.2 mmol/L).

| Intervention | CVD Risk (%, 95%CI) | CVD RD (%, 95%CI) | NNT | All-cause mortality Risk (%, 95%CI) | All-cause mortality RD (%, 95%CI) | NNT | ASCVD Risk (%, 95%CI) | ASCVD RD (%, 95%CI) | NNT |
|---|---|---|---|---|---|---|---|---|---|
| **Natural course** | 18.6 (17.7 to 19.9) | | | 25.6 (24.7 to 27.4) | | | 17.7 (16.9 to 19.0) | | |
| **Feasible interventions** | 16.4 (15.6 to 18.5) | -2.1 (-3.1 to -0.7) | 124 | 22.8 (22.0 to 25.3) | -2.8 (-4.1 to -1.7) | 100 | 15.3 (14.4 to 17.3) | -2.4 (-3.3 to -1.1) | 112 |
| **Subgroup analyses** | | | | | | | | | |
| **Sex** | | | | | | | | | |
| Women | 14.6 (11.4 to 17.2) | -2.3 (-4.9 to 0.1) | 122 | 19.6 (15.6 to 23.1) | -2.7 (-5.9 to 0.3) | 109 | 13.4 (10.5 to 15.8) | -2.7 (-5.1 to 0.1) | 108 |
| Men | 19.3 (16.7 to 22.5) | -0.7 (-3.2 to 2.2) | 376 | 29.0 (25.6 to 32.6) | -1.5 (-4.2 to 1.2) | 176 | 18.3 (15.6 to 21.3) | -0.9 (-3.5 to 1.7) | 275 |
| **BMI (kg/m²)** | | | | | | | | | |
| < 24 | 12.3 (10.2 to 15.4) | -2.7 (-5.0 to 0.6) | 98 | 17.8 (14.4 to 21.2) | -4.4 (-7.5 to -0.9) | 65 | 11.8 (9.9 to 14.9) | -2.5 (-4.7 to 1.1) | 108 |
| ≥ 24 | 22.7 (19.7 to 25.2) | -0.9 (-4.1 to 1.5) | 312 | 30.2 (26.0 to 34.1) | -0.6 (-3.3 to 3.5) | 467 | 20.7 (18.0 to 23.2) | -1.7 (-4.8 to 0.3) | 154 |
| **Smoking status** | | | | | | | | | |
| Yes | 21.0 (17.8 to 27.3) | -3.2 (-5.7 to 1.0) | 81 | 32.7 (26.5 to 41.6) | -2.9 (-7.8 to 2.8) | 95 | 20.1 (16.9 to 25.7) | -3.4 (-5.6 to 0.5) | 79 |
| No | 16.4 (14.1 to 19.2) | -0.9 (-2.6 to 1.5) | 306 | 21.5 (18.7 to 24.2) | -1.9 (-4.3 to 0.6) | 148 | 15.3 (13.1 to 18.2) | -1.2 (-3.0 to 1.3) | 235 |
| **Antihypertensive medication** | | | | | | | | | |
| Yes | 35.8 (26.3 to 41.8) | 2.9 (-2.2 to 9.3) | -57 | 48.2 (36.3 to 56.4) | 6.0 (1.5 to 12.2) | -31 | 34.3 (24.9 to 39.6) | 2.2 (-3.1 to 8.4) | -73 |
| No | 16.3 (14.2 to 19.1) | -1.9 (-3.2 to 0.7) | 142 | 21.7 (19.2 to 24.5) | -3.5 (-5.4 to -1.2) | 82 | 15.1 (13.3 to 17.7) | -2.2 (-3.5 to 0.2) | 127 |
| **Sensitivity analyses** | | | | | | | | | |
| Reordering time-varying variables | 16.5 (15.6 to 18.5) | -2.1 (-3.1 to -0.7) | 127 | 22.8 (22.0 to 25.3) | -2.8 (-4.1 to -1.7) | 100 | 15.3 (14.4 to 17.3) | -2.4 (-3.2 to -1.1) | 114 |
| **Positive control analyses** | | | | | | | | | |
| Per 1 mmol/L LDL-C reduction | 15.5 (12.9 to 18.2) | -3.1 (-6.1 to -0.9) | 79 | 21.2 (17.9 to 24.8) | -4.5 (-8.2 to -2.3) | 58 | 13.8 (11.4 to 16.5) | -3.9 (-6.7 to -1.8) | 65 |

**Fig. 5 | Forest plots of the estimated effects under feasible interventions.** Forest plot of the estimated parametric g-formula risk (%), risk difference (RD, %), and restricted mean event-free based number needed to treat (NNT) to prevent one cardiovascular, all-cause mortality, and atherosclerotic cardiovascular event under natural course and feasible interventions (unless stated) after 29 years of follow-up from the Chinese Multi-provincial Cohort Study 1992-2020. Note: [†] Feasible treat-to-target cholesterol-lowering intervention, defined as 80% of eligible participants receiving the intervention at the follow-up examination over the study period, where the treat-to-target cholesterol-lowering intervention is based on cholesterol-lowering targets recommended by the Chinese Society of Cardiology in 2020 on LDL-C and non-HDL-C levels, i.e., for participants with diabetes at high cardiovascular risk, lower the LDL-C to < 1.8 mmol/L (70 mg/dL, i.e., a fixed level drawn from a uniform distribution with a upper bound of 1.8 mmol/L) or LDL-C reduction to > 50% from baseline whichever is the lowest and non-HDL-C to < 2.6 mmol/L (100 mg/dL, i.e., a fixed level drawn from a uniform distribution with a upper bound of 2.6 mmol/L); for participants without diabetes who are at moderate-to-high cardiovascular risk lower the LDL-C to < 2.6 mmol/L (100 mg/dL, i.e., a fixed level drawn from a uniform distribution with a upper bound of 2.6 mmol/L) and non-HDL-C to < 3.4 mmol/L (130 mg/dL, i.e., a fixed level drawn from a uniform distribution with a upper bound of 3.4 mmol/L); for participants at low cardiovascular risk, lower LDL-C to < 3.4 mmol/L (130 mg/dL, i.e., a fixed level drawn from a uniform distribution with a upper bound of 3.4 mmol/L) and a non-HDL-C < 4.2 mmol/L (160 mg/dL, i.e., a fixed level drawn from a uniform distribution with a upper bound of 4.2 mmol/L).

**Table 3 | Estimated parametric g-formula risk (%), risk difference (RD, %), risk ratio (RR), restricted mean event-free time (RMET, years), and restricted mean event-free based number needed to treat (NNT) to prevent one cancer death under natural course, treat-to-target interventions recommended by the 2020 Chinese Society of Cardiology and per 1 mmol/L LDL-C reduction after 29 years of follow-up from the Chinese Multi-provincial Cohort Study 1992-2020**

| Interventions | Cancer mortality | | | | | |
|---|---|---|---|---|---|---|
| | Risk (%, 95% CI) | RD (%, 95% CI) | RR (95% CI) | RMET (years) | NNT | Average % intervention |
| Natural course | 8.6 (7.8 to 9.3) | Reference | Reference | 28.4 | Reference | 0.0 |
| Treat-to-target interventions[a] | 7.2 (3.8 to 11.3) | −1.4 (−4.7 to 2.4) | 0.84 (0.45 to 1.27) | 28.5 | 294 | 29% |
| Feasible interventions[b] | 7.2 (4.0 to 11.3) | −1.3 (−4.5 to 2.4) | 0.85 (0.46 to 1.27) | 28.5 | 310 | 27% |
| Per 1 mmol/L LDL-C reduction | 8.3 (4.5 to 14.3) | -0.3 (−4.1 to 5.7) | 0.86 (0.52 to 1.67) | 28.4 | 1218 | 100% |

Note: [a] Treat-to-target cholesterol-lowering intervention is based on cholesterol-lowering targets recommended by the Chinese Society of Cardiology in 2020 on LDL-C and non-HDL-C levels, i.e., for participants with diabetes at high cardiovascular risk, lower the LDL-C to <1.8 mmol/L (70 mg/dL, i.e., a fixed level drawn from a uniform distribution with an upper bound of 1.8 mmol/L) or LDL-C reduction to > 50% from baseline whichever is the lowest and non-HDL-C to <2.6 mmol/L (100 mg/dL, i.e., a fixed level drawn from a uniform distribution with an upper bound of 2.6 mmol/L); for participants without diabetes who are at moderate-to-high cardiovascular risk lower the LDL-C to < 2.6 mmol/L (100 mg/dL, i.e., a fixed level drawn from a uniform distribution with an upper bound of 2.6 mmol/L) and non-HDL-C to < 3.4 mmol/L (130 mg/dL, i.e., a fixed level drawn from a uniform distribution with a upper bound of 3.4 mmol/L); for participants at low cardiovascular risk, lower LDL-C to < 3.4 mmol/L (130 mg/dL, i.e., a fixed level drawn from a uniform distribution with a upper bound of 3.4 mmol/L) and a non-HDL-C < 4.2 mmol/L (160 mg/dL, i.e., a fixed level drawn from a uniform distribution with a upper bound of 4.2 mmol/L).
[b]Feasible treat-to-target cholesterol-lowering intervention, defined as 80% of eligible participants receiving the intervention at the follow-up examination over the study period.

**Table 4 | Estimated treat-to-target cholesterol-lowering[a] effects on cardiovascular disease (CVD), all-cause mortality, and atherosclerotic cardiovascular disease (ASCVD) using parametric g-formula estimators and doubly-robust estimators**

| Outcomes | Parametric g-formula estimator | Doubly-robust estimator |
|---|---|---|
| CVD | | |
| Risk ratio (95% CI) | 0.88 (0.82 to 0.96) | 0.90 (0.88 to 0.93) |
| Absolute risk reduction (%, 95% CI) | −2.3 (−3.4 to −0.8) | −2.1 (−3.4 to −0.7) |
| All-cause mortality | | |
| Risk ratio (95% CI) | 0.88 (0.84 to 0.93) | 0.91 (0.88 to 0.94) |
| Absolute risk reduction (%, 95% CI) | −3.0 (−4.3 to −1.8) | −2.5 (−4.1 to −0.9) |
| ASCVD | | |
| Risk ratio (95% CI) | 0.85 (0.81 to 0.93) | 0.88 (0.85 to 0.90) |
| Absolute risk reduction (%, 95% CI) | −2.6 (−3.5 to −1.2) | −2.6 (−4.3 to −0.9) |

Note: [a]Treat-to-target cholesterol-lowering intervention is based on cholesterol-lowering targets recommended by the Chinese Society of Cardiology in 2020 on LDL-C and non-HDL-C levels, i.e., for participants with diabetes at high cardiovascular risk, lower the LDL-C to <1.8 mmol/L (70 mg/dL, i.e., a fixed level drawn from a uniform distribution with a upper bound of 1.8 mmol/L) or LDL-C reduction to > 50% from baseline whichever is the lowest and non-HDL-C to <2.6 mmol/L (100 mg/dL, i.e., a fixed level drawn from a uniform distribution with a upper bound of 2.6 mmol/L); for participants without diabetes who are at moderate-to-high cardiovascular risk lower the LDL-C to < 2.6 mmol/L (100 mg/dL, i.e., a fixed level drawn from a uniform distribution with a upper bound of 2.6 mmol/L) and non-HDL-C to < 3.4 mmol/L (130 mg/dL, i.e., a fixed level drawn from a uniform distribution with a upper bound of 3.4 mmol/L); for participants at low cardiovascular risk, lower LDL-C to < 3.4 mmol/L (130 mg/dL, i.e., a fixed level drawn from a uniform distribution with a upper bound of 3.4 mmol/L) and a non-HDL-C < 4.2 mmol/L (160 mg/dL, i.e., a fixed level drawn from a uniform distribution with a upper bound of 4.2 mmol/L).

no interventions. Moreover, the higher the adherence rate of the cholesterol-lowering interventions, the greater the benefit.

## Methods
We first briefly describe the protocol of the target trial and then emulate it as closely as possible using longitudinal data from CMCS from 1992 to the end of 2020, as depicted in Table 1.

### Study participants
This study initially included 5966 participants aged 35 years or older from the Chinese Multi-provincial Cohort Study recruited at the baseline during 1992-1993 ($W_0$)[49,50]. Informed consent was obtained from all participants, and this study was approved by the ethics committee of Beijing Anzhen Hospital, Capital Medical University. Briefly, we excluded 231 participants, including 106 with prevalent CVD and 125 with TG ≥4.52 mmol/L (400 mg/dL) at $W_0$, leaving 5735 participants available in the final analysis, as detailed in Fig. 1. Of these, participants free of CVD were actively invited to participate in follow-up examinations in 2002 ($W_1$, 2011 participants from the Beijing area), 2007 ($W_2$, 5353 participants from the Beijing, Tianjin, Heilongjiang, Liaoning, and Sichuan provinces), 2012 ($W_3$, 1739 participants from the Beijing area). Of these, 5413 had one follow-up examination, 2182 had two, and 1508 had three. Information on demographics, lifestyles, and medical history was collected using a standardized questionnaire modified based on the WHO-MONICA protocol[51], with clinical measurements tested in the laboratory during the baseline and follow-up examinations. All participants were actively followed up for the onset of CVD events or any non-CVD deaths every 1 to 2 years, supplemented via the local disease surveillance systems. All reported CVD events and non-CVD deaths were adjudicated by a panel of physicians. Consequently, the loss to follow-up rate was relatively low[50], thus not materially impacting the cholesterol-lowering effects.

### Protocol for target trial
**Eligibility criteria.** Participants are ≥ 35 years of age between 1 January 1992 and 31 December 1993, with in-person examination, no prevalent CVD, and triglyceride (TG) < 4.52 mmol/L (400 mg/dL) at baseline.

**Treatment strategies and assignment.** Eligible participants are assigned to one of the following treatment strategies when risk-based conditions are met at the follow-up examination over the study period: (1) Natural course of no cholesterol-lowering interventions, i.e., no interventions were implemented over the study period. (2) Long-term target-to-target cholesterol-lowering interventions, in which we chose the cholesterol-lowering target levels following the CSC recommendations based on predicted 10-year and lifetime risk-based LDL-C and non-HDL-C targets for all eligible participants over the study period[2]. Specifically, for participants with diabetes at high cardiovascular risk, lower the LDL-C to < 1.8 mol/L (70 mg/dL, i.e., a fixed level drawn from a uniform distribution with an upper bound of 1.8 mmol/L) or LDL-C reduction to > 50% from baseline whichever is the lowest and non-HDL-C to < 2.6 mmol/L (100 mg/dL, i.e., a fixed level drawn from a uniform distribution with an upper bound of 2.6 mmol/L); for participants without diabetes who are at moderate-to-high cardiovascular risk lower the LDL-C to < 2.6 mmol/L (100 mg/dL, i.e., a fixed level drawn from a

uniform distribution with an upper bound of 2.6 mmol/L) and non-HDL-C to < 3.4 mmol/L (130 mg/dL, i.e., a fixed level drawn from a uniform distribution with a upper bound of 3.4 mmol/L); for participants at low cardiovascular risk, lower LDL-C to < 3.4 mmol/L (130 mg/dL, i.e., a fixed level drawn from a uniform distribution with an upper bound of 3.4 mmol/L) and a non-HDL-C < 4.2 mmol/L (160 mg/dL, i.e., a fixed level drawn from a uniform distribution with an upper bound of 4.2 mmol/L). (3) Feasible treat-to-target cholesterol-lowering intervention, in which 80% of eligible participants receive intervention at each follow-up examination over the study to mimic clinical practice according to a pragmatic cholesterol-lowering trial[52]. All interventions are compared with the "natural course" of no interventions. To appropriately implement the treat-to-target intervention, a regular assessment of participants' risk profiles must be conducted, assuming all participants have a biannual or annual follow-up examination during the study period. We stopped the cholesterol-lowering interventions immediately after the cholesterol-lowering targets were met.

**Outcomes.** The primary outcomes are incident cardiovascular events (CVD, defined as cardiovascular death, acute myocardial infarction, stroke, cardiac attest, heart failure, and coronary revascularization) and all-cause mortality. The secondary outcome is ASCVD, defined as a composite endpoint including acute coronary and ischemic stroke events or excluding hemorrhagic stroke from CVD events. Any non-CVD deaths, including cancer deaths and other deaths precluding the occurrence of outcomes of interest over the study period, are defined as competing events.

**Follow-up.** Eligible participants are followed from the baseline until the date of recording CVD, non-CVD deaths, loss to follow-up, 29 years after the baseline, or administrative end of follow-up on 31 December 2020, whichever occurs first.

### Target trial emulation

**Eligibility criteria.** We strictly applied all eligibility criteria to CMCS participants. Moreover, we limited the study participants to those with in-person examinations over the study period, aiming to achieve a high adherence rate.

**Treatment strategies and assignment.** Unlike the static intervention implemented at the baseline, treat-to-target interventions are time-involving and participants can receive the corresponding intervention when the risk-based conditions are met during follow-up in-person examinations over the study period. Thus, we emulated the treatment strategies by transforming the follow-up time into a one-year unit and initiating the intervention when the conditions were met. We defined the start of the follow-up period (i.e., time zero) as the initiation time of the intervention when risk-based conditions were met. We stopped the cholesterol-lowering interventions immediately after the cholesterol-lowering targets were met.

**Causal contrasts.** The per-protocol effects of being assigned and fully adhered to treat-to-target cholesterol-lowering interventions over the study period *versus* the "natural course" without any interventions are quantified by the absolute risk reduction (ARR), rate ratios (RR), and the restricted mean event-free time (RMET, i.e., the area above the cumulative curve or under the survival curve from time zero to a specific time point)-based number needed to treat (NNT)[36] to prevent one extra CVD or all-cause death till 29 years of follow-up.

**Statistical analysis.** We estimated the cumulative risk of developing CVD, all-cause mortality, and ASCVD under the treat-to-target intervention and the "natural course" via the parametric g-formula[24,26]. Specifically, we used pooled logistic or linear regression models to fit the time-varying confounders, outcome, competing events, and

exposure separately. Then, we estimated the cumulative risk by standardizing the cardiovascular risk resulting from the hypothetical treat-to-target interventions using Monte Carlo draws of confounders from the fitted exposure model. Based on the estimated cumulative risk, we calculated ARR, RR, and the RMET-based NNT till 29 years follow-up to quantify the long-term effect of the treat-to-target interventions on CVD, all-cause death, and ASCVD from a public health perspective. To implement parametric g-formula, an outline is described as follows,

1. Construct a set of regression models using the pooled person-time data
   a. Fit regression models for all time-varying confounders, using cholesterol levels, previous intervention, confounder histories, and time as predictors.
   b. Fit regression models for cholesterol levels, using previous cholesterol levels and intervention, confounder histories, and time as predictors.
   c. Fit pooled logistic regression models for CVD/all-cause mortality, using previous cholesterol levels and intervention, confounder histories, and cubic polynomial function of time in a one-year unit as predictors among surviving and remaining uncensored participants.
   d. Fit pooled logistic regression models for competing events with previous cholesterol levels and intervention, confounder histories, and cubic polynomial function of time in a one-year unit as predictors among surviving and remaining uncensored participants.
2. Use a Monte Carlo simulation to generate 10,000 individuals under the treat-to-target cholesterol-lowering intervention. For each individual,
   a. Randomly sample values of baseline characteristics, including confounders and cholesterol levels, with replacements from the original study participants.
   b. Iteratively draw the time-varying confounders from the fitted time-varying confounder models.
   c. Draw cholesterol levels from the fitted cholesterol levels regression models and assign the risk-based treat-to-target cholesterol-lowering interventions to eligible participants when conditions are met over the study period.
   d. Estimate the confounder-specific cumulative incidence risk of CVD and all-cause mortality using pooled logistic outcome regression models after accounting for competing events.
3. Calculate the mean cumulative incidence risk of developing CVD and all-cause mortality under the treat-to-target intervention at the end of a 29-year follow-up, with the NNT estimated based on the RMET.
4. Repeat Steps 1-3 in 500 bootstrap samples to obtain the percentile 95% confidence intervals (CI).

To adjust for potential confounders in Step 1, we specified a set of cardiovascular risk factors based on previous studies[41,53]. Figure 6 depicts the assumed causal relationships of treat-to-target cholesterol-lowering interventions with CVD, all-cause mortality, and ASCVD at each follow-up point in the presence of competing events. Specifically, we selected age at baseline and sex as time-fixed confounders, and cholesterol levels of LDL-C, non-HDL-C, cholesterol-lowering treatment, body mass index (BMI), systolic blood pressure (SBP), triglyceride (TG), smoking status, diabetes status, cardiovascular risk status, and use of antihypertensive drugs are considered time-varying confounders, of which cholesterol levels of LDL-C, non-HDL-C, cholesterol-lowering treatment were regarded as treat-to-target intervention variables. When the time-varying variables were not measured, we imputed missing values using the most recently measured values of the in-person follow-up examination since the last measurement over the study period (i.e., the last observation carried forward) as recommended by Hernán, et al.[54] and the use of predicted

## a Target trial specification

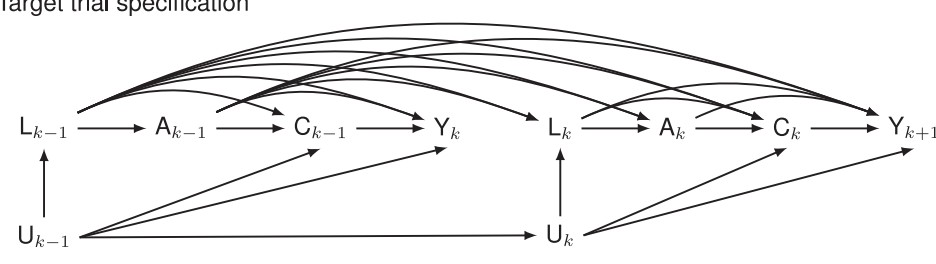

## b Target trial emulation

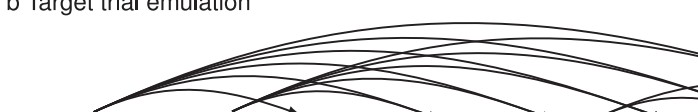
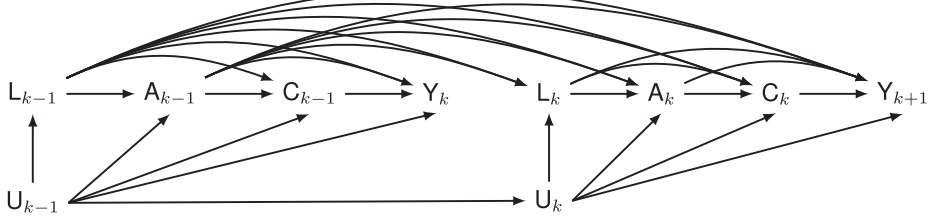

**Fig. 6 | Causal diagrams of target trial specification and emulation.** Causal graphs illustrate the effects of the treat-to-target cholesterol-lowering interventions (*A*) on cardiovascular disease, all-cause mortality, and atherosclerotic cardiovascular disease (*Y*) in the presence of competing events (*C*) using the sequentially randomized trial (the upper panel, i.e., target trial specification, **a** in which the intervention strategy $A_k$ only depends on prior cardiovascular risk profiles ($L_k$), cholesterol levels and intervention history ($A_{k-1}$) at each time $k (k = 0, 1, ..., 28)$ and the hypothetical pragmatic trial (the bottom panel, i.e., target trial emulation, **b** in which $A_k$ depends on prior $L_k$, $A_{k-1}$, and the potential unmeasured confounders $U_k$. $C_{k-1}$ and $Y_k$ indicate the competing events at time interval $k - 1$ and the outcome of interest at time interval $k$, respectively.

values from the generalized linear mixed model failed to appropriately reflect their corresponding levels at each follow-up visit (Supplementary Fig. 1). To predict the joint distribution of time-varying varying variables, the lifestyle factors (e.g., smoking status), metabolic factors (e.g., BMI, SBP, TG, non-HDL-C, LDL-C), cardiovascular risk status, and medications (e.g., antihypertensive and cholesterol-lowering drugs) are successively modeled, with details of model forms presented in Supplementary Table 1. Moreover, we considered deaths from non-cardiovascular events competing risks to obtain the total effects of treat-to-target cholesterol-lowering interventions and avoid the interpretation of direct effect by censoring these non-cardiovascular events in an unrealistic and counterfactual scenario where all non-cardiovascular deaths were eliminated alongside other possible forms of censoring[55]. The same steps are also implemented for the secondary outcome of ASCVD.

Furthermore, the NNT for preventing one cardiovascular event is computed as the RMET under the "natural course" divided by the difference in RMETs between the treat-to-target intervention and the "natural course" till 29 years of follow-up[36]. To probe potential model misspecification, the standardized cumulative risk curves and the parametric g-formula estimated risk curves under the natural course are compared. Any discrepancies between these risk curves could indicate the model misspecification over the study period. Finally, the average proportion of participants who would have to receive treat-to-target interventions at any time during the study is estimated to explore the potential extrapolation of the intervention.

To identify vulnerable participants for whom the treat-to-target interventions may confer more benefit, subgroup analyses by sex (men and women), body mass index (< 24 and ≥ 24 kg/m²), smoking status (yes and no), and antihypertensive drugs (yes and no) at baseline are also conducted separately. Lastly, the sensitive analysis and positive/negative control analyses are conducted using the same data to validate the estimated effects as follows: (1) reorder the time-varying variables in Step 1; (2) vary the adherence rate of the treat-to-target

intervention over the study period from 20% to 70% by 10%; and (3) replicate the constant ~21% cardiovascular risk reduction with statin therapies per 1 mmol/L (39 mg/dL) reduction in plasma LDL-C levels as positive control analysis and the null association of cholesterol-lowering interventions[16] with cancer deaths as negative outcome analysis[56,57]. We conducted all analyses using R 4.3.0 using gfoRmula[40] and nnt[36] packages.

### Reporting summary
Further information on research design is available in the Nature Portfolio Reporting Summary linked to this article.

## Data availability
The data are currently private due to privacy laws, ethical restrictions, and confidentiality agreements. For specific academic requests to access these data, please contact the corresponding author, Professor Jing Liu (jingliu@ccmu.edu.cn), with a detailed research request and may be required to sign a data use agreement to protect participant confidentiality. Professor Jing Liu will respond within 4 weeks. Source data will be provided once the research requests are approved.

## Code availability
The sample code used for this study is publicly available on Github: http://github.com/yangzhao98/dynamicIntervention (https://doi.org/10.5281/zenodo.13844279).

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

## Acknowledgements
This work was supported by the National Natural Science Foundation of China (grant 12226005), the National Key Research and Development Program of China (grant 2022YFC3602501), the Beijing Municipal Commission of Health (grant 2021-7), and the Beijing Anzhen Hospital High-Level Research Funding (grant 2024AZB2002). The funding source had no involvement in the study design, in the collection, analysis, and interpretation of data, in the writing of the report, or in the decision to submit the article for publication. The authors also thank the investigators from all collaborative centers who contributed to collecting data and appreciate the research participants in this study.

## Author contributions
Z.Y. and J.L. conceived the idea and design of the study. Y.Q., P.Z., Y.M.H., Z.Y.W, W.L.Z., and J.L. contributed to data collection and management. Z.Y. analyzed the data and wrote the first draft. Z.Y., Q.J.D, Y.C.H., N.Y., L.Z.H., P.P.J., P.Z., Y.M.H, Z.Y.W., W.L.Z., Y.Q., and J.L. contributed to the interpretation of the findings. Z.Y., Q.J.D, Y.C.H., N.Y., L.Z.H., P.P.J., P.Z., Y.M.H., Z.Y.W., W.L.Z., Y.Q., and J.L. critically revised the paper for intellectual content and approved the final version of the manuscript.

## Competing interests
The Authors declare no competing interests.
