## [Transparent Peer Review file · Nature Communications]

Effectiveness of treat-to-target cholesterol-lowering interventions on cardiovascular disease and all-cause mortality risk in the community-dwelling population: a target trial emulation

Corresponding Author: Professor Jing Liu

Version 0:

Reviewer comments:

Reviewer #1

(Remarks to the Author)

Major comments:

- 1) The exact definition of the intervention is not mentioned in the abstract or early sections of the paper or in the related tables and figures. I would suggest adding it early on even though it is rather long. All tables and figures, certainly Table 1, should also have a footnote to describe the intervention fully.
- 2) I would also suggest using a more descriptive label for the intervention (as opposed to dynamic). My suggestion is to call it treat-to-target.
- 3) As the authors have mentioned, the consistency assumption may not hold for this type of intervention. However, this is not mentioned in the Discussion section. Please add as a limitation.
- 4) Relatedly, I would suggest changing the design of the positive control analysis in such a way that the intervention is receiving statins as opposed to modeling the impact of an equivalent reduction in LDL-cholesterol.
- 5) I would also suggest conducting a negative control outcome analysis. For example, using cancer incidence or mortality.
- 6) The point estimates reported in Figure 4 often fall near the end of the uncertainty interval. This happens when using a fairly low number of simulations using exponential models. I suggest the authors report the median estimate for the quantity of interest across simulations instead.
- 7) It would be great if the authors reported a causal graph in the appendix and used that to select the variables. As is, the logic behind the choice of variables is not clear. It seems that a few major potential confounders are not included: alcohol use, diet quality, physical activity and history of major chronic diseases such as CKD, chronic liver disease, COPD. Please clarify this in the limitations.
- 8) The second part of the Methods, starting from page 28 under the title 'target trial emulation' is fairly repetitive. I strongly suggest combining this with the previous section and removing the repetitions.
- 9) It is rather unclear how the authors handled missing data across waves or the potential for informative censoring when participants were not invited or did not show up on follow-up waves. The details need to be added to the Methods.
- 10) Based on Extended Data Figure 1, it seems that collectively the regression models underestimate all three outcomes later in the follow-up. I suggest examining and reporting the simulated vs. observed estimates for all covariates across the waves. This may show which models may be mis-specified leading to this underestimation.

Minor comments

- 11) Abstract: the last sentence mentions a change in NNT from 279 to 131. However, the 279 estimate has not been reported earlier on. Which outcome does this relate to?
- 12) Page 5: spell out RMET in first use.
- 13) Page 8 second line: report duration of follow-up for the absolute risk reduction in ref #31.
- 14) Page 9, first sentence of second paragraph: randomized should be changed to non-randomized
- 15) Table 2: add calendar years to each wave.
- 16) Table 2: add footnote for 'Risk stratification' to specify which risk prediction model was used (and cite) and which thresholds were used to define low, intermediate, and high risk categories.
- 17) Figure 2: add (%) for unit to cumulative risk axis.

18) Methods, page 26 item 1.d. report which functional form and power was used to model time and was that in years, months, ..?

19) Extended Table 3: Average % intervention estimates can/should be rounded to the nearest digit.

Reviewer #2

(Remarks to the Author)

I appreciate the opportunity to review the manuscript “Dynamic cholesterol-lowering interventions reduce cardiovascular disease and 1 all-cause mortality risk in the community-dwelling population: a target trial emulation using data from a 29-year cohort study”. The authors use the parametric g-formula to study hypothetical interventions to reduce cholesterol using dynamic treatment regimes based on recent clinical guidelines. I believe this work is relevant, and their implementation of the target trial framework and g-formula methods is clear. However, there are some sections of the target trial protocol and design that require work to improve the clearness and readability. Please find below some questions and suggestions:

Methods

Target trial specification

Eligibility criteria:

1. The authors include participants aged 35 years or older between January 1, 1992 and December 31, 2020. Why did the authors include the whole study period vs. the first wave of recruitment?

2. The criteria of having 1 follow-up examination after baseline is not intuitive because in the ideal RCT they would not recruit, randomize, and assign participants to the different treatment arms at time 1 and only include those who returned at time 2. So it should not be in the target trial specification, unless they imagine that the randomization and treatment assignment starts in wave 2 instead. However, I do believe they should discuss why in practice they have to make this study decision in the target trial emulation section.

3. It is not clear if the eligibility criteria is reassessed at each wave of data. If so, are they emulating sequential nested trials? See ref: <https://www.ncbi.nlm.nih.gov/pmc/articles/PMC4832051/>. Because they are using the parametric g-formula, I think this is not the case?. But I think the authors are confusing the eligibility criteria with the criteria to get the intervention. They should keep in mind that the randomization starts at baseline but under any of the intervention arms, the treatment strategy says change x to x1 whenever this criteria is met over the follow-up. Please clarify this throughout the sections and in the table 1.

Treatment Strategies:

4. According to Figure 2, the authors have 3 hypothetical treatment strategies plus the natural course. They should be clear about this in this section, and specify them in Table 1. Currently, they are not specifying the interventions as expected. See the following work for reference: <https://www.ncbi.nlm.nih.gov/pmc/articles/PMC2786249/>.

5. On the first intervention (line 456 – 457) “with diabetes at high cardiovascular risk, lower the LDL-C to <1.8 mmol/L (70 mg/dL) or LDL-C reduction to >50% from baseline”. How did they decide to do either <1.8 or >50%? What was the cut-off point?

6. The “feasible intervention” is not clear, how did they specify adherence to 80%? Would this be translated as an arm where: if a person adheres for 80% of their follow-up then do nothing afterwards?

7. Are the diabetes diagnosis and cardiovascular risk measured at every time? Is the condition on the same time-point as the intervention?

Statistical analysis:

8. Did the authors look at the mean difference over time between the observed values and predicted values under the natural course for all the time-varying covariates included in the model? Since they include most continuous covariates as linear terms, I think this can induce model misspecification. Please show these as plots in the appendix.

Target trial emulation:

9. The authors should describe in more detail the CMCS, specifically how many waves of data is available, what are the years in between waves, and the summary of the range of time between visits for all participants. Because it seems that the outcome data comes from different sources, the authors should specify what variables are collected within the waves, and how they harmonized it with the outcome data. Considering that the parametric g-formula is sensitive to model misspecifications, understanding this information of the data is crucial.

10. Do they have the specific date/year of death or do they only know if a participant has died by wave t+1?

11. Previous studies that consider multiple sources of data have modeled the visit process as well, this is relevant because the intervention can only happen if they come to the study visit, because the time between visits probably varies by each individual, and because the time between study visits is substantially long (5 years), they should consider this to improve the modeling. See <https://link.springer.com/article/10.1007/s10654-020-00694-5>.

12. How did they account for missing data?

13. Why the sample size increase from W1 to W2?

14. The authors should define how they constructed the “risk stratification” variable that is used for the interventions.

15. Did the authors constructed a dataset with 1-year intervals or only rows as waves?
16. Is the 29-year risk is presented as a probability? Why not a %? Also, is it a cause-specific cumulative incidence, so that when looking at CVD and ASCVD, they account for the probability of all cause-mortality?
17. I suggest authors to also include the Risk ratios for the main analysis.
18. Please refer to <https://www.ncbi.nlm.nih.gov/pmc/articles/PMC7811594/> when the total effect is discussed in terms of competing events.

Discussion:

19. The authors should discuss the nuances of the consistency assumption, and how this intervention is a representation of a weighted average of all the strategies (including taking statins?) that are present in the observed data. See: <https://link.springer.com/article/10.1007/s10654-020-00694-5>.
20. In line 216, the authors say: "To the best of our knowledge, this is the first randomized (hypothetical) target trial showing that long-term risk-based dynamic cholesterol-lowering interventions could...". Please rephrase to: To the best of our knowledge, this is the first target trial emulation of hypothetical dynamic cholesterol-lowering.... showing that long term...

Abstract:

21. Since the target trial emulates the study design that answers an estimands, but not the estimands per se, I would remove the "the preventive effects of" from the following sentence: "This target trial emulated the preventive effects of dynamic cholesterol-lowering 14 interventions on cardiovascular events using data from the Chinese Multi-provincial 15 Cohort Study."

Reviewer #3

(Remarks to the Author)

In this manuscript, the authors study a cohort of nearly 6,000 individuals in a Chinese community dwelling situation from a primary prevention perspective and investigate the potential impact on achieving lower LDL-C levels on long term CV and all-cause mortality outcomes followed for 29 years. The data confirms effective long term lowering of the risk of these outcomes. The findings are interesting and confirm the broader concept of the benefits of lipid lowering and more recent studies that show greater benefit on longer term follow up. The authors should consider a number of points.

1. The main concern is how does this really advance the field. The literature already has considerable data showing the benefits of lipid lowering, that benefit has durability on long term follow up and there is even data supporting a potential legacy effect. What is important about this findings are that they are generated from randomised clinical trials and lack the limitations of emulation.
2. As outlined by the authors their findings are completely dependent on emulation and what goes in to the model. It cannot exclude residual confounding and in my opinion is similarly challenged, as are other approaches highlighted by the authors – simulation has strengths, but considerable limitations as well.
3. I'm not sure the size of this cohort is particularly big. In fact, very few individuals are followed up at wave 1 and 3. It would seem that this would further challenge the assumptions implicit within the model.

Version 1:

Reviewer comments:

Reviewer #1

(Remarks to the Author)

I think the paper is substantially improved and is much clearer now.

I would like to suggest that the authors add a bit more detail on the method of imputing missing data across time. As is, it is not clear if data were simply carried forward from the last observation or if a (multiple) imputation approach was used.

Reviewer #2

(Remarks to the Author)

The authors have addressed all my comments and suggestions, the current manuscript reads clearly. I have a few minor comments:

- The current title of the manuscript is set up as a conclusion. To maintain objectivity and encourage critical engagement of readers, consider revising the title to a more neutral and descriptive format that reflects the study's aim/design rather than interpreting the results. Otherwise it can lead to misinterpretations, considering that the interpretation requires understanding the limitations and potential sources of bias in the analysis.

- Page 4, lines 86 – 91. The sentence "Lastly, randomized controlled trials, particularly sequentially randomized experiments randomly assigned treat-to-target cholesterol lowering" is a bit lengthy but doesn't convey as much information as it could. I think it would make a stronger point to say that we mostly care about per protocol effects, rather than the ITT (probably what most RCT's report), but for that we need to account for time-varying confounding feedback. While the idea of

sequential randomized experiments helps convey the need to remove the confounder feedback loops, it does not necessarily mean that the interest is on long-term effects.

- Line 297, author's start a sentence with "Finally" that is followed by another sentence with "Fourth".

- In the limitations section, the authors need to include that the intervals between waves are very long, and although the g-formula is set to "intervene" on a yearly basis, these are effectively happening whenever data is updated (every 5 – 10 years). I think this is hinted in line 305 where they mention more regular repeated measurements, but they should be more explicit of the limitation.

- Furthermore, participants who were not from Beijing and attended wave 0 and 2 would have a 15-year gap between measurements. Authors must explain why this decision is more reasonable than restricting analysis to those participants from Beijing area only, and what assumptions are they willing to make with this decision (in terms of carrying forward data for such a long time).

Reviewer #3

(Remarks to the Author)

The authors are thanked for their thoughtful responses. This reviewer remains concerned that the same limitations that were raised are what they are. There are fundamental limitations to simulation, they don't overcome the limitations of real world clinical trials and the other reviewers have highlighted other factors that are real - simulation assumes consistency, non confounding and perfection - this clinical trialist can humbly aspire to work in such a world.

the authors should tone down the suggestion that all approaches to LDL-C lowering, including diet, will produce such findings. The reality is that dietary effects on LDL-C are modest and it is misleading of the authors to suggest that their simulated findings would be achieved by people simply adopting a diet.

Point-by-Point Responses to Comments from Reviewers on Manuscript

NCOMMS-24-03531-T

Thank you for your careful review of our manuscript and many insightful comments. Considering all your comments, we have revised our manuscript, which led to a much better exposition of our work. Our point-by-point responses to your comments are given below, with your original comments copied in *Italics* for your convenience. Any material from the revised manuscript is in **bold**, surrounded by quotation marks.

Comments:

Reviewer #1 (Remarks to the Author):

Major comments:

1. The exact definition of the intervention is not mentioned in the abstract or early sections of the paper or in the related tables and figures. I would suggest adding it early on even though it is rather long. All tables and figures, certainly Table 1, should also have a footnote to describe the intervention fully.

Response: Thank you for your comments. We have added the definition of “treat-to-target cholesterol-lowering intervention” in the Abstract (page 2, lines 6-11 in the Manuscript) and the Method (page 22, lines 1-25) sections. We have also added footnotes to all tables (pages 17 & 19 & 33-34 & 36 & 39 & 41-43) and figures (pages 20-21).

2. I would also suggest using a more descriptive label for the intervention (as opposed to dynamic). My suggestion is to call it treat-to-target.

Response: Thank you very much for this invaluable suggestion. “Treat-to-target” explicitly describes the meaning of the intervention. We have updated the dynamic cholesterol-lowering interventions throughout the manuscript to “treat-to-target interventions”.

3. As the authors have mentioned, the consistency assumption may not hold for this type of intervention. However, this is not mentioned in the Discussion section. Please add as a limitation.

Response: Thank you for your comments. We added the following text in the Discussion-Limitation section to explain the possibility and consequences of violating the consistency assumption (pages 10-11, lines 25-29 & 1-4), “**Third, the interpretation of our causal estimates also relies on the well-defined interventions and consistency assumption – the counterfactual outcome under the observed intervention value is equal to the observed outcome for each subject. However, we didn’t explain how cholesterol levels would be lowered, which means the observed protective estimates can be results from either lifestyle or pharmaceutical interventions or any other approaches, making the consistency assumption dubious in the current study Hernán ¹. Therefore, our estimates should be interpreted as an overall effect of various LDL-C lowering strategies in the CMCS population ². Further studies with detailed assessments of cholesterol-lowering interventions remain needed.**”

4. Relatedly, I would suggest changing the design of the positive control analysis in such a way that the intervention is receiving statins as opposed to modeling the impact of an equivalent reduction in LDL-cholesterol.

Response: Thank you for your comments. According to your suggestion, we emulated a target trial for the positive control analysis using moderate statin therapies recommended by the Chinese Society of Cardiology for Primary Prevention of Cardiovascular Diseases ³. To this end, we first extracted the LDL-cholesterol effects of various moderate statin therapies from previous studies ^{4,5} (**Table A1**), based on which the LDL-C levels were expected to reduce by approximately 40% compared with the pretreatment levels for participants with baseline LDL-C ≥ 1.8 mmol/L. We found consistent cholesterol-lowering effects on all-cause mortality (rate ratio: 0.83;

95%CI: 0.72 to 0.98) and atherosclerotic cardiovascular disease (0.86; 0.71 to 1.01), with a slight attenuated effect on cardiovascular disease (0.89; 0.74 to 1.06). We have added these results in the **Extended Data Tables 2-3** (pages 35 & 36 in the Manuscript). Furthermore, we kept the per 1 mmol/L LDL-C lowering effects to align with the meta-analyzed effects presented by Cholesterol Treatment Trialists' (CTT) Collaborators as an alternative positive control analysis ⁶, with which a relative cardiovascular risk reduction of 21% per 1 mmol/L LDL-C lowering has been well-acknowledged.

Table A1. Percentage of LDL-cholesterol reduction (%) compared with the pretreatment levels of moderate statin therapies for various drugs

Drugs	Dosage (mg)	Mean (%)	Standard derivation (%)
Atovastatin	10	35.5	10.6
	20	41.4	13.5
Fluvastatin	80	26.0	9.0
Lovastatin	40	30.3	11.0
	60	34.5	11.7
Pitavastatin	2	39.0	14.6
	4	44.0	14.2
Pravastatin	40	30.0	11.2
	80	33.0	13.0
Rosuvastatin	5	38.8	13.2
	10	44.1	12.5
Simvastatin	20	33.0	10.4
	40	38.9	14.0

5. I would also suggest conducting a negative control outcome analysis. For example, using cancer incidence or mortality.

Response: Thank you for your suggestions. We emulated the treat-to-target cholesterol-lowering interventions on cancer mortality as a negative control outcome analysis. As expected, no association between cholesterol-lowering and cancer mortality was observed, further supporting our conclusions. We have added this as a part of sensitivity analyses (pages 6-7, lines 9-11; page 29, lines 9-10 in the manuscript) and supplemented the results as **Extended Data Table 6** (page 42).

6. The point estimates reported in Figure 4 often fall near the end of the uncertainty interval. This happens when using a fairly low number of simulations using exponential models. I suggest the authors report the median estimate for the quantity of interest across simulations instead.

Response: Thank you for your suggestions. Since the lipid-lowering effects are approximately normally distributed based on 500 bootstrapping samples, the median and mean estimates are almost identical, with slight variations largely due to random errors. Nevertheless, we have specified the method used for constructing the 95% confidence interval in the Method section (page 26, lines 22-23), “**Repeated Steps 1-3 in 500 bootstrap samples to obtain the percentile 95% confidence intervals (CI).**” Moreover, we have revised **Figure 3&4** accordingly to illustrate the treat-to-target cholesterol-lowering effects.

7. It would be great if the authors reported a causal graph in the appendix and used that to select the variables. As is, the logic behind the choice of variables is not clear. It seems that a few major potential confounders are not included: alcohol use, diet quality, physical activity and history of major chronic diseases such as CKD, chronic liver disease, COPD. Please clarify this in the limitations.

Response: Thank you for your suggestions. We have added the following causal graphs (**Extended Data Figure 1**) in the Method section on pages 26-27 to illustrate the effects of the treat-to-target cholesterol-lowering interventions.

Target trial specification

Target trial emulation

Extended Data Figure 1. Causal graphs illustrate the effects of the treat-to-target cholesterol-lowering interventions (A , joint intervention strategy, including LDL-C, non-HDL-C, and cholesterol-lowering treatments) on cardiovascular disease, all-cause mortality, and atherosclerotic cardiovascular disease (Y) in the presence of competing events (C) using the sequentially randomized trial (the upper panel, i.e., target trial specification) in which the intervention strategy A_k only depends on the baseline characteristics (i.e., age at baseline and sex, which are omitted from the graph for simplicity), time-varying cardiovascular risk profiles L_k (i.e., body mass index, systolic pressure, triglyceride, smoking status, diabetes status, use of antihypertensive drugs), and intervention history (A_{k-1}) at each time k ($k = 0, 1, \dots, 28$) and the hypothetical pragmatic trial emulation (the bottom panel, i.e., target trial emulation) in which A_k depends on prior L_k , A_{k-1} , and the potential unmeasured confounders U_k .

Based on causal diagrams, we selected age at baseline and sex as time-fixed confounders and cholesterol levels of LDL-C, non-HDL-C, cholesterol-lowering treatments, body mass index (BMI), systolic blood pressure (SBP), triglyceride (TG), smoking status, diabetes status, cardiovascular risk stratification status, use of antihypertensive drugs as time-varying confounders, which are usually considered the modified cardiovascular risk factors^{7,8}. We have highlighted these points by adding the following text in the Statistical Analysis section, **“To adjust for potential confounders in Step 1, we specified a set of cardiovascular risk factors based on**

previous studies^{7,8}. Extended Data Figure 1 depicts the assumed causal relationships of treat-to-target cholesterol-lowering interventions with CVD, all-cause mortality, and ASCVD at each follow-up point in the presence of competing events. Specifically, we selected age at baseline and sex as time-fixed confounders, and cholesterol levels of LDL-C, non-HDL-C, cholesterol-lowering treatment, body mass index (BMI), systolic blood pressure (SBP), triglyceride (TG), smoking status, diabetes status, cardiovascular risk status, and use of antihypertensive drugs are considered time-varying confounders, of which cholesterol levels of LDL-C, non-HDL-C, cholesterol-lowering treatment were regarded as treat-to-target intervention variables.”

We didn't include some other potential confounders of the history of major chronic diseases, such as CKD, chronic liver disease, COPD, alcohol use, physical activity, and dietary habits. Failing to adjust for these confounders may bias our estimates apart from the null. However, except for CKD, these potential confounders are generally associated with CVD through their effect on classic risk factors and are not considered in CVD risk assessment⁷. In recent years, CKD has been increasingly recognized as an important contributor to CVD risk, but it has not been included in Chinese guidelines^{3,9} until 2020. Although data for CKD and liver disease was not available for the 1992 baseline, the prevalence of CKD 3-4 and elevated liver function (defined as alanine aminotransferase [ALT] and/or aspartate aminotransferase [AST] is elevated ≥ 3 times the upper limit of normal [ALT: 50 U/L for male and 40 U/L for female in Chinese adults¹⁰] and combined with elevated total bilirubin [≥ 26.0 umol/L for male and ≥ 21 umol/L for female]¹¹) were only 3.2% and 0.3% in 2002, respectively. Therefore, their impacts on our results would be minimal. We have added the above text to the Discussion-Limitation section (page 10, lines 4-17 in the Manuscript).

8. The second part of the Methods, starting from page 28 under the title ‘target trial emulation’ is fairly repetitive. I strongly suggest combining this with the previous section and removing the repetitions.

Response: Thank you for your suggestions. We combined the target trial specification and target trial emulation, particularly the causal contrast and statistical analyses sections, as outlined in the Method section on pages 22-28 in the Manuscript.

9. It is rather unclear how the authors handled missing data across waves or the potential for informative censoring when participants were not invited or did not show up on follow-up waves. The details need to be added to the Methods.

Response: Thank you for your comments. To emulate the dynamic treat-to-target cholesterol-lowering interventions, we discrete the follow-up time into a one-year unit, with the missing values of time-varying confoundings and intervention variables imputed using the most recently measured values of the in-person follow-up examination since the last measurement, as recommended by Hernán, et al. ¹². We have added these points in the Method section (page 27, lines 6-9 in the Manuscript).

10. Based on Extended Data Figure 1, it seems that collectively the regression models underestimate all three outcomes later in the follow-up. I suggest examining and reporting the simulated vs. observed estimates for all covariates across the waves. This may show which models may be mis-specified leading to this underestimation.

Response: Thank you for your comments. We have updated the regression model for each time-varying confounder, which leads to better model fitting, as depicted in **Extended Data Figure 2**. Moreover, we triangulated our results using the sequentially doubly robust estimators for the treat-to-target effects, which are robust to the model misspecification in the longitudinal settings ¹³⁻¹⁵. As expected, similar treat-to-target cholesterol-lowering effects on CVD, all-cause mortality, and ASCVD were noted regarding relative or absolute risk reduction, enhancing the credibility of

our results. We have supplemented these results as **Extended Data Table 7** in the manuscript (page 43).

Minor comments

11. Abstract: the last sentence mentions a change in NNT from 279 to 131. However, the 279 estimate has not been reported earlier on. Which outcome does this relate to?

Response: Thank you for your comments. We apologize for the confusion, and we have revised the sentence: “**Increasing adherence rate from 20% to 70% was associated with a halved NNT for preventing one CVD, supporting treat-to-target cholesterol-lowering interventions in the community-dwelling population** (page 2, lines 18-21).”

12. Page 5: spell out RMET in first use.

Response: Thank you for your comments. We spelled out the RMET as the restricted mean of event-free time in its first use on page 5, lines 25-26 in the Manuscript.

13. Page 8 second line: report duration of follow-up for the absolute risk reduction in ref #31.

Response: Thank you for your comments. We have provided the follow-up information of statin trials on page 8, lines 12-13 in the Manuscript, “**Similarly, our findings were consistent with the pooled analysis from the US Preventive Services Task Force, showing a 1.3% absolute risk reduction after a mean follow-up of 3.3 years (ranging from 6 months to 6 years of follow-up)¹⁶, enhancing the credibility of our results**”.

14. Page 9, first sentence of second paragraph: randomized should be changed to non-randomized.

Response: Thank you for your comments. We have changed the “**randomized (hypothetical) target trial**” to “**pragmatic target trials,**” as advocated by Hernan and Robins ^{17,18}.

15. Table 2: add calendar years to each wave.

Response: Thank you for your comments. We added each wave's calendar years to **Table 2** on pages 18-19 in the Manuscript.

16. Table 2: add footnote for ‘Risk stratification’ to specify which risk prediction model was used (and cite) and which thresholds were used to define low, intermediate, and high risk categories.

Response: We have explained the risk assessment algorithm used in the current study by adding the following text as a footnote for **Table 2** on page 19, “**The cardiovascular risk stratification was assessed following the ASCVD and CVD risk algorithms recommended by the Chinese guideline for the primary prevention of cardiovascular disease ³, in which the 10-year risk stratification of total cardiovascular risk is the same as the atherosclerotic cardiovascular disease in most cases. Briefly, a 3-step evaluation procedure was employed: (1) participants with diabetes (aged ≥ 40 years) and LDL-C ≥ 4.9 mmol/L (or total cholesterol [TC] ≥ 7.2 mmol/L) were directly classified at high risk; (2) participants who do not meet procedure (1), sex-specific 10-year ASCVD risk assessment algorithms, including LDL-C or TC levels, hypertension, smoking status, HDL-C, and age $\geq 45/55$ years (male/female), were used ¹⁹, based on which 10-year ASCVD risk $< 5\%$, $5\%–9\%$, and $\geq 10\%$ were defined as low, intermediate, and high risk, respectively; (3) for those intermediate-risk participants aged < 55 years, the lifetime risk of cardiovascular disease were assessed, in which participants with ≥ 2 following risk factors are defined at high risk: a) systolic blood pressure ≥ 160 mmHg or diastolic blood pressure ≥ 100** ”

mmHg; b) non-HDL-C ≥ 5.2 mmol/L (200 mg/dL); c) HDL-C < 1.0 mmol/L (40 mg/dL); d) body mass index (BMI) ≥ 28 kg/m²; and e) smoking.”

17. Figure 2: add (%) for unit to cumulative risk axis.

Response: Thank you for your comments. We have revised **Figure 2** accordingly.

18. Methods, page 26 item 1.d. report which functional form and power was used to model time and was that in years, months, ..?

Response: Thank you for your comments. We have explained the cubic polynomial function form used for a time by adding the following text on page 25, lines 27-30, “**Fit pooled logistic regression models for competing events with previous cholesterol levels and intervention, confounder histories, and cubic polynomial function of time in a one-year unit as predictors among surviving and remaining uncensored participants** (page 26, lines 1-4).”

19. Extended Table 3: Average % intervention estimates can/should be rounded to the nearest digit.

Response: Thank you for your comments. We have revised the **Extended Data Table 3** on pages 35-36.

Comments from Reviewer #2 (Remarks to the Author):

I appreciate the opportunity to review the manuscript “Dynamic cholesterol-lowering interventions reduce cardiovascular disease and 1 all-cause mortality risk in the community-dwelling population: a target trial emulation using data from a 29-year cohort study”. The authors use the parametric g-formula to study hypothetical interventions to reduce cholesterol using dynamic treatment regimes based on recent clinical guidelines. I believe this work is relevant, and their implementation of the target trial framework and g-formula methods is clear. However, there are some sections of the target trial protocol and design that require work to improve the clearness and readability. Please find below some questions and suggestions:

Methods

Target trial specification

Eligibility criteria:

1. The authors include participants aged 35 years or older between January 1, 1992 and December 31, 2005. Why did the authors include the whole study period vs. the first wave of recruitment?

Response: Thank you for your comments. We apologize for the confusion. We limited our study participants to those aged 35 years or older recruited during 1992-1993, with in-person examination, no prevalent CVD, and no missing values of cholesterol levels ^{20,21}. We have revised these points in the Method Section on pages 22-23, lines 30&1-2.

2. The criteria of having 1 follow-up examination after baseline is not intuitive because in the ideal RCT they would not recruit, randomize, and assign participants to the different treatment arms at time 1 and only include those who returned at time 2. So it should not be in the target trial specification, unless they imagine that the

randomization and treatment assignment starts in wave 2 instead. However, I do believe they should discuss why in practice they have to make this study decision in the target trial emulation section.

Response: Thank you for your comments. We first provided causal diagrams (**Data Extended Figure 1, on page 44**) to illustrate the target trial specification and how we emulated such a trial using longitudinal data from the Chinese Multi-provincial Cohort Study. We restricted our analyses to participants who participated in the in-person examination during the recruitment period²⁰⁻²² to ensure a high adherence rate²³. In the target trial emulation section, we have explained these points by adding the following text on page 24, lines 15-17 in the Manuscript: **“We strictly applied all eligibility criteria to CMCS participants. Moreover, we limited the study participants to those with in-person examinations over the study period, aiming to achieve a high adherence rate.”**

Moreover, we explained how to implement the dynamic intervention at each follow-up examination over the study in the *Treatment strategies and assignment* section on pages 24-25, lines 22-30 in the Manuscript, **“Unlike the static intervention implemented at the baseline, treat-to-target interventions are dynamic, and participants can receive the corresponding intervention when the risk-based conditions are met at multiple follow-up examinations over the study period. Thus, we emulated the treatment strategies by transforming the follow-up time into a one-year unit and initiating the intervention when the conditions were met. We defined the start of the follow-up period (i.e., time zero) as the initiation time of the intervention when risk-based conditions were met. We stopped the cholesterol-lowering interventions immediately after the cholesterol-lowering targets were met.”**

3. It is not clear if the eligibility criteria is reassessed at each wave of data. If so, are they emulating sequential nested trials? See

ref: <https://www.ncbi.nlm.nih.gov/pmc/articles/PMC4832051/>. Because they are using the parametric g-formula, I think this is not the case?. But I think the authors are confusing the eligibility criteria with the criteria to get the intervention. They should keep in mind that the randomization starts at baseline but under any of the intervention arms, the treatment strategy says change x to $x1$ whenever this criteria is met over the follow-up. Please clarify this throughout the sections and in the table 1.

Response: Thank you for your invaluable comments and clarification. Yes, the eligibility criteria for recruiting study participants were only assessed at the baseline, while the dynamic cholesterol-lowering interventions can be initiated when the risk-based conditions are met at the follow-up examinations over the study period, with time zero defined as the initiation time of the intervention ²³. We have revised **Table 1** (pages 16-17) and the **Method section** (page 24, lines 25-29), as detailed in **Comment #2**.

Treatment Strategies:

4. According to Figure 2, the authors have 3 hypothetical treatment strategies plus the natural course. They should be clear about this in this section, and specify them in Table 1. Currently, they are not specifying the interventions as expected. See the following work for

reference: <https://www.ncbi.nlm.nih.gov/pmc/articles/PMC2786249/>.

Response: Thank you for your comments. We have specified the three hypothetical cholesterol-lowering treatment strategies in the Method-*Treatment strategies and assignment* section on page 23, lines 4-28, "**Eligible participants are assigned to one of the following treatment strategies when risk-based conditions are met over the follow-up period: (1) Natural course of no cholesterol-lowering interventions, i.e., no interventions were implemented over the study period. (2) Long-term target-to-target cholesterol-lowering interventions, in which we chose the cholesterol-lowering target levels following the CSC recommendations based on**

predicted 10-year and lifetime risk-based LDL-C and non-HDL-C targets for all eligible participants over the study period ³. Specifically, for participants with diabetes at high cardiovascular risk, lower the LDL-C to <1.8 mmol/L (70 mg/dL, i.e., a fixed level drawn from a uniform distribution with an upper bound of 1.8 mmol/L) or LDL-C reduction to >50% from baseline whichever is the lowest and non-HDL-C to <2.6 mmol/L (100 mg/dL, i.e., a fixed level drawn from a uniform distribution with an upper bound of 2.6 mmol/L); for participants without diabetes who are at moderate-to-high cardiovascular risk lower the LDL-C to <2.6 mmol/L (100 mg/dL, i.e., a fixed level drawn from a uniform distribution with an upper bound of 2.6 mmol/L) and non-HDL-C to <3.4 mmol/L (130 mg/dL, i.e., a fixed level drawn from a uniform distribution with an upper bound of 3.4 mmol/L); for participants at low cardiovascular risk, lower LDL-C to <3.4 mmol/L (130 mg/dL, i.e., a fixed level drawn from a uniform distribution with an upper bound of 3.4 mmol/L) and a non-HDL-C <4.2 mmol/L (160 mg/dL, i.e., a fixed level drawn from a uniform distribution with an upper bound of 4.2 mmol/L). (3) Feasible treat-to-target cholesterol-lowering intervention, in which 80% of eligible participants receive intervention at each follow-up examination to mimic clinical practice according to a pragmatic cholesterol-lowering trial ²⁴.”

5. *On the first intervention (line 456 – 457) “with diabetes at high cardiovascular risk, lower the LDL-C to <1.8 mmol/L (70 mg/dL) or LDL-C reduction to >50% from baseline”. How did they decide to do either <1.8 or >50%? What was the cut-off point?*

Response: Thank you for your comments. In our analyses, to mimic the target LDL-C level recommended by the Chinese Society of Cardiology for primary prevention of cardiovascular disease ³, we decided the LDL-C level based on a fixed level drawn from a uniform distribution with an upper bound of 1.8 mmol/L or LDL-C reduction

to >50% from baseline, whichever is the lowest, for those participants at high cardiovascular risk, as stated in **Comment #4**.

6. *The “feasible intervention” is not clear, how did they specify adherence to 80%? Would this be translated as an arm where: if a person adheres for 80% of their follow-up then do nothing afterwards?*

Response: Thank you for your comments. The adherence rate of 80% means that 80% of eligible participants, when the risk-based conditions were met at follow-up examination, would receive the treat-to-target cholesterol-lowering intervention, mimicking the clinical practice with a feasible adherence rate, as noticed in the pragmatic cholesterol-lowering trial ²⁴. We clarified this point in the Treatment Strategies section on page 23, lines 25-28 in the Manuscript as follows, “(3) **Feasible treat-to-target cholesterol-lowering intervention, in which 80% of eligible participants receive intervention at each follow-up examination to mimic clinical practice according to a pragmatic cholesterol-lowering trial** ²⁴.”

7. *Are the diabetes diagnosis and cardiovascular risk measured at every time? Is the condition on the same time-point as the intervention?*

Response: Yes. Since eligible participants were followed up and re-examined over the study period, diabetes diagnosis and cardiovascular risk profiles were repeatedly measured. Consequently, dynamic treat-to-target cholesterol-lowering interventions can be initiated when the risk-based conditions are met during the follow-up examinations. We have added causal graphs, **Extended Data Figure 1**, as detailed in **Comment #2**, to illustrate the time-varying status of diabetes and cardiovascular risk profiles over the study period.

Statistical analysis:

8. *Did the authors look at the mean difference over time between the observed values and predicted values under the natural course for all the time-varying covariates*

included in the model? Since they include most continuous covariates as linear terms, I think this can induce model misspecification. Please show these as plots in the appendix.

Response: Thank you for your comments. We have updated the regression model for each time-varying confounder, which leads to better model fitting, as depicted in **Extended Data Figure 2** on page 46 of the Manuscript. Moreover, we triangulated our results using the sequentially doubly robust estimators for the treat-to-target effects (**Extended Data Table 7** on page 44 in the Manuscript), which are robust to the model misspecification for the longitudinal data¹³⁻¹⁵. Please refer to our response to **Comment #10** from **Reviewer 1** on pages 7-8 in this response letter.

Target trial emulation:

9. The authors should describe in more detail the CMCS, specifically how many waves of data is available, what are the years in between waves, and the summary of the range of time between visits for all participants. Because it seems that the outcome data comes from different sources, the authors should specify what variables are collected within the waves, and how they harmonized it with the outcome data. Considering that the parametric g-formula is sensitive to model misspecifications, understanding this information of the data is crucial.

Response: Thank you for your comments. To illustrate, we have detailed the longitudinal data of study participants from the CMCS in the Method-Study participants' section on page 22, lines 7-27, as follows, “**This study initially included 5,966 participants aged 35 years or older from the Chinese Multi-provincial Cohort Study, an ongoing community-based prospective study in China, recruited at the baseline during 1992-1993 (W_0). Details of the CMCS have been previously published^{20,21}. This study was approved by the ethics committee of Beijing Anzhen Hospital, Capital Medical University. Briefly, we excluded 231 participants, including 106 with prevalent CVD, and 125 with TG ≥ 4.52 mmol/L (400 mg/dL) at W_0 , leaving 5,735 participants available in the final analysis, as**

detailed in Figure 1. Of these, participants free of CVD were actively invited to participate in follow-up examinations in 2002 (W_1 , 2011 from the Beijing area), 2007 (W_2 , 5,353 from the Beijing, Tianjin, Heilongjiang, Liaoning, and Sichuan provinces), 2012 (W_3 , 1,739 from the Beijing area). Of these, 5,413 had one follow-up examination, 2,182 had two, and 1,508 had three. Information on demographics, lifestyles, and medical history was collected using a standardized questionnaire modified based on the WHO-MONICA protocol ²², with clinical measurements tested in the laboratory, during the baseline and follow-up examinations. All participants were actively followed up for the onset of CVD events or any non-CVD deaths every 1 to 2 years, supplemented *via* the local disease surveillance systems. All reported CVD events and non-CVD deaths were adjudicated by a panel of physicians. Consequently, the loss to follow-up rate was relatively low ²¹, thus not materially impacting the cholesterol-lowering effects.”

10. Do they have the specific date/year of death or do they only know if a participant has died by wave $t+1$?

Response: Yes, we have participants' exact date of death by actively following up every 1 to 2 years, supplemented via the local disease surveillance systems. We have explained these points in the Method section on page 22, lines 22-25, “**All participants were actively followed up for the onset of CVD events or any non-CVD deaths every 1 to 2 years, supplemented *via* the local disease surveillance systems. All reported CVD events and non-CVD deaths were adjudicated by a panel of physicians.**”

11. Previous studies that consider multiple sources of data have modeled the visit process as well, this is relevant because the intervention can only happen if they come to the study visit, because the time between visits probably varies by each individual, and because the time between study visits is substantially long (5 years), they should

consider this to improve the modeling.

See <https://link.springer.com/article/10.1007/s10654-020-00694-5>.

Response: Yes, it is true. The initiation of the dynamic treat-to-target cholesterol-lowering intervention can only happen at each follow-up visit when the risk-based conditions are met. Thus, our analyses discretized the follow-up time into a one-year unit (i.e., person-time data), with the missing values of time-varying variables being imputed using the most recently measured in-person follow-up examination since the last measurement, as recommended by Hernán, et al.¹². We initiated the cholesterol-lowering intervention when risk-based conditions were met over the study period and used the parametric g-formula to estimate the cumulative risk of developing CVD, all-cause mortality, and ASCVD under various hypothetical interventions. Specifically, we used pooled logistic or linear regression models to fit the time-varying confounders, outcome, competing events, and exposure separately. Then, we estimated the cumulative risk by standardizing the cardiovascular risk resulting from the hypothetical dynamic interventions using Monte Carlo draws of confounders from the fitted exposure model, as detailed in the **Statistical analysis section** on pages 25-26.

12. How did they account for missing data?

Response: Thank you for your comments. We removed participants with missing values for key variables at the baseline, such as LDL-C levels, and imputed the missing values of time-varying confounders during follow-up using the last observation carried forward method. We have explained this point in the Method section on page 27, lines 6-9 in the Manuscript by adding the following text: “**When the time-varying variables were not measured, we imputed missing values using the most recently measured values of the in-person follow-up examination since the last measurement over the study period, as recommended by Hernán, et al.¹²”**

13. *Why the sample size increase from W1 to W2?*

Response: Thank you for your comments. To illustrate, we first detailed the longitudinal data from study participants used in this study, as stated in **Comment #9** on **page 16**. Briefly, only participants from the Beijing area in the original Chinese Multi-provincial Cohort Study were invited to participate in the follow-up examinations in 2002 (W₁) and 2012 (W₃), with additional participants from the Beijing, Tianjin, Heilongjiang, Liaoning, and Sichuan provinces participating the follow-up examination in 2007 (W₂). Of these, 5,413 had one follow-up examination, 2,182 had two, and 1,508 had three. Thus, the sample size increases from W₁ to W₂.

14. *The authors should define how they constructed the “risk stratification” variable that is used for the interventions.*

Response: Thank you for your comments. We have explained the risk assessment algorithm used in the current study by adding the following text as a footnote for Table 2 on page 19 in the Manuscript, “**The cardiovascular risk stratification was assessed following the ASCVD and CVD risk algorithms recommended by the Chinese guideline for the primary prevention of cardiovascular disease²⁵, in which the 10-year risk stratification of total cardiovascular risk is the same as the atherosclerotic cardiovascular disease in most cases. Briefly, a 3-step evaluation procedure was employed: (1) participants with diabetes (aged ≥ 40 years) and LDL-C ≥ 4.9 mmol/L (or total cholesterol [TC] ≥ 7.2 mmol/L) were directly classified at high risk; (2) participants who do not meet procedure (1), sex-specific 10-year ASCVD risk assessment algorithms, including LDL-C or TC levels, hypertension, smoking status, HDL-C, and age $\geq 45/55$ years (male/female), were used¹⁹, based on which 10-year ASCVD risk $< 5\%$, $5\%–9\%$, and $\geq 10\%$ were defined as low, intermediate, and high risk, respectively; (3) for those intermediate-risk participants aged < 55 years, the lifetime risk of cardiovascular disease were assessed, in which participants with ≥ 2 following risk factors are defined at high risk: a) systolic blood pressure**

≥160 mmHg or diastolic blood pressure ≥100 mmHg; b) non-HDL-C ≥5.2 mmol/L (200 mg/dL); c) HDL-C <1.0 mmol/L (40 mg/dL); d) body mass index (BMI) ≥28kg/m²; and e) smoking.”

15. Did the authors constructed a dataset with 1-year intervals or only rows as waves?

Response: Yes, we constructed the data in a one-year unit to emulate the dynamic treat-to-target cholesterol-lowering intervention and initiated the intervention when the conditions were met. We defined the start of the follow-up period (i.e., time zero) as the initiation time of the intervention when risk-based conditions were met, as detailed in **Comment #2**.

16. Is the 29-year risk is presented as a probability? Why not a %? Also, is it a cause-specific cumulative incidence, so that when looking at CVD and ASCVD, they account for the probability of all cause-mortality?

Response: Thank you for your comments. Yes, it is a probability. We have added % for all reported risks. Moreover, in our analyses, competing events from any non-CVD deaths or hemorrhagic strokes were considered when estimating the cumulative risk of developing CVD and ASCVD. Thus, the probability of all cause-mortality (i.e., overall survival) was also accounted for. We explained this point in the Statistical analysis section on page 27, lines 13-18 in the Manuscript: **“Moreover, we considered deaths from non-cardiovascular events competing risks to obtain the total effects of treat-to-target cholesterol-lowering interventions and avoid the interpretation of direct effect by censoring these non-cardiovascular events in an unrealistic and counterfactual scenario where all non-cardiovascular deaths were eliminated alongside other possible forms of censoring²⁶. The same steps are also implemented for the secondary outcome of ASCVD.”**

17. I suggest authors to also include the Risk ratios for the main analysis.

Response: Thank you for your comments. We have added the risk ratios in the main analysis in the Result-Estimated effects of treat-to-target interventions section on pages 5-6 in the Manuscript beyond the **Extended Data Tables 3&5**.

18. Please refer to <https://www.ncbi.nlm.nih.gov/pmc/articles/PMC7811594/> when the total effect is discussed in terms of competing events.

Response: Thank you for your comments. We have interpreted the 29-year cumulative risk of developing CVD and ASCVD as total effects in the presence of competing events, adhering to the study you recommended, as stated in **Comment #16**.

Discussion:

19. The authors should discuss the nuances of the consistency assumption, and how this intervention is a representation of a weighted average of all the strategies (including taking statins?) that are present in the observed data.

See <https://link.springer.com/article/10.1007/s10654-020-00694-5>.

Response: Thank you for your comments. We have added discussion on this issue in the Limitation section on pages 10-11, lines 25-29&1-5 in the Manuscript: “**Third, the interpretation of our causal estimates also relies on the well-defined interventions and consistency assumption – the counterfactual outcome under the observed intervention value is equal to the observed outcome for each subject. However, we didn’t explain how cholesterol levels would be lowered, which means the observed protective estimates can be results from either lifestyle or pharmaceutical interventions or any other approaches, making the consistency assumption dubious in the current study ¹. Therefore, our estimates should be interpreted as an overall effect of various LDL-C lowering strategies in the CMCS population ². Further studies with detailed assessments of cholesterol-lowering interventions remain needed.**”

20. In line 216, the authors say: “To the best of our knowledge, this is the first randomized (hypothetical) target trial showing that long-term risk-based dynamic cholesterol-lowering interventions could...”. Please rephrase to: To the best of our knowledge, this is the first target trial emulation of hypothetical dynamic cholesterol-lowering.... showing that long term...

Response: Thank you for your comments. We have revised the sentence as follows on page 9, lines 15-20, “**To the best of our knowledge, this is the first pragmatic target trial emulating the hypothetical dynamic cholesterol-lowering interventions and showing that long-term treat-to-target risk-based cholesterol-lowering interventions could exert an equivalent protective effect concerning the statin therapeutic trials on cardiovascular risk in the community-dwelling population, particularly when maintaining the adherence rate at a high level (e.g., 70%-80%).**”

Abstract:

21. Since the target trial emulates the study design that answers an estimands, but not the estimands per se, I would remove the “the preventive effects of” from the following sentence: “This target trial emulated the preventive effects of dynamic cholesterol-lowering 14 interventions on cardiovascular events using data from the Chinese Multi-provincial 15 Cohort Study.

Response: Thank you for your comments. We have revised this sentence in the Abstract on page 2, lines 2-4: “**This target trial emulated long-term treat-to-target cholesterol-lowering interventions on cardiovascular events using longitudinal data from the Chinese Multi-provincial Cohort Study.**”

Reviewer #3 (Remarks to the Author):

In this manuscript, the authors study a cohort of nearly 6,000 individuals in a Chinese community dwelling situation from a primary prevention perspective and investigate the potential impact on achieving lower LDL-C levels on long term CV and all-cause mortality outcomes followed for 29 years. The data confirms effective long term lowering of the risk of these outcomes. The findings are interesting and confirm the broader concept of the benefits of lipid lowering and more recent studies that show greater benefit on longer term follow up. The authors should consider a number of points.

1. The main concern is how does this really advance the field. The literature already has considerable data showing the benefits of lipid lowering, that benefit has durability on long term follow up and there is even data supporting a potential legacy effect. What is important about this findings are that they are generated from randomised clinical trials and lack the limitations of emulation.

Response: We sincerely appreciate your time to offer invaluable insights on our paper. We agree with you that several post-trials showed the potential legacy effects of lipid-lowering (within trials) treatments during an extended follow-up of trial cohorts²⁷⁻³². However, using post-randomization data after the end of the trial breaks the randomization at baseline and makes them no longer randomized comparisons^{28,33}. Consequently, the legacy effects from such randomized trials were typically analyzed as observational³⁴, making their interpretations intangible. Specifically, there are two main reasons, as pointed out by Toh and Hernán³⁵:

- 1) They fail to distinguish the within-trial cholesterol-lowering effects from post-trial intervention effects;
- 2) They are vulnerable to post-randomization confounding (e.g., post-randomization prognostic factors not only affected by prior treatment but also affected future treatment decisions) and selection bias (e.g., differential nonadherence and loss to follow-up rates between groups).

These make most trials with baseline randomization optimal for detecting small treatment benefits during the short trial period ¹⁶, but not for studying the long-term effects of sustained clinical interventions in eligible patients and primary care settings ³⁴. Moreover, it has been well-recognized that the clinical benefits of lipid-lowering treatment evolve over time, with a smaller reduction in cardiovascular risk during the first few years than in subsequent years ³⁶. This further complicates the clinical interpretation of hazard ratios in existing lipid-lowering trials ³⁷⁻³⁹.

In such a context, to the best of our knowledge, this is the first pragmatic target trial showing that long-term risk-based treat-to-target cholesterol-lowering interventions could exert an equivalent protective effect concerning the statin therapeutic trials on cardiovascular risk in the community-dwelling population, particularly when maintaining the adherence rate at a high level (e.g., 70%-80%), after accounting for time-varying confounders and selection bias under several untestable and often plausible assumptions ^{17,18}. The consistent results of sensitivity, positive/negative control analyses, and the subsequent doubly robust estimators enhance the credibility of our results. Moreover, using the restricted mean event-free time-based number needed to treat provides clinically meaningful treat-to-target effects of long-term cholesterol-lowering interventions ³⁹ in primary care settings. Lastly, unlike most lipid-lowering trials showing “potential legacy effects,” our findings provide first-hand evidence of dynamic and long-term risk-based cholesterol-lowering effects on preventing CVD, ASCVD, and all-cause mortality in the community-dwelling populations.

2. As outlined by the authors their findings are completely dependent on emulation and what goes in to the model. It cannot exclude residual confounding and in my opinion is similarly challenged, as are other approaches highlighted by the authors – simulation has strengths, but considerable limitations as well.

Response: Thank you for your comments. Yes, a central challenge of specifying and emulating a target trial for dynamic interventions *via* parametric g-formula is the strong assumption of no unmeasured confounding depending on the past intervention and covariate history over the study period, which is often not guaranteed to hold in an observational study ⁴⁰. For example, we didn't include some other potential confounders of the history of major chronic diseases, such as CKD, chronic liver disease, COPD, alcohol use, physical activity, and dietary habits. Failing to adjust for these confounders may bias our estimates apart from the null. However, except for CKD, these covariates are generally associated with CVD through their effect on classic risk factors and are not considered in CVD risk assessment ⁷. In recent years, CKD has been increasingly recognized as an important contributor to CVD risk, but it has not been included in Chinese guidelines ^{3,9} until 2020. Although data for CKD and liver disease was not available for the 1992 baseline, the prevalence of CKD 3-4 and elevated liver function (defined as alanine aminotransferase [ALT] and/or aspartate aminotransferase [AST] is elevated ≥ 3 times the upper limit of normal [ALT: 50 U/L for male and 40 U/L for female in Chinese adults; AST: 40 U/L for male and 35 U/L for female in Chinese adults¹⁰] and combined with elevated total bilirubin [26.0 umol/L for male and 21 umol/L for female]¹¹) were only 3.2% and 0.3% in 2002, respectively. Therefore, their impacts on our results would be minimal. We have added the above text to the Discussion-Limitation section on page 10, lines 4-17.

3. I'm not sure the size of this cohort is particularly big. In fact, very few individuals are followed up at wave 1 and 3. It would seem that this would further challenge the assumptions implicit within the model.

Response: Thank you for your comments. Yes, the sample size of our study is not very large, although it is comparable with most lipid-lowering trials with a median sample size of 4,509 participants (ranging from 1,255 to 20,536) ⁴¹. However, the consistent estimates of benefits of dynamic and long-term treat-to-target cholesterol-

lowering interventions on CVD, ASCVD, and all-cause mortality across various analyses in our study, supplemented by positive (i.e., mimicking statins therapy effects) and negative (i.e., mimicking no effects of cholesterol-lowering interventions on cancer mortality) examples suggest our study had adequate power by using 4,509 participants for the target trial emulation.

Additionally, analyses of sequentially doubly robust estimators using cross-fitting to allow for flexible machine learning regression methodology and avoid parametric model misspecification while maintaining valid statistical inference were conducted^{13,14}. The consistent dynamic and long-term treat-to-target effects of cholesterol-lowering on CVD, all-cause mortality, and ASCVD further validate our findings, as detailed in the **Extended Data Table 7** on page 43 in the Manuscript.

Nevertheless, we highlighted these points in the Limitation section on page 11, lines 14-18: **“Finally, our findings might have been underpowered and imprecise due to the relatively small sample size⁴². A large-scale longitudinal study with more regular repeated measurements of traditional cardiovascular risk profiles, lifestyle, medication, and disease history (e.g., major chronic diseases) would further triangulate our results.”**

Reference:

1. Hernán, M.A. Invited commentary: hypothetical interventions to define causal effects—afterthought or prerequisite? *American journal of epidemiology* **162**, 618–620; discussion 621–612 (2005).
2. Hernán, M.A. & VanderWeele, T.J. Compound treatments and transportability of causal inference. *Epidemiology (Cambridge, Mass.)* **22**, 368–377 (2011).
3. Chinese Society of Cardiology of Chinese Medical Association, C. D. P. a. R. C. o. C. A. o. R. M., Cardiovascular Disease Committee of Chinese Association of Gerontology and Geriatrics, Thrombosis Prevention and Treatment Committee of Chinese Medical Doctor Association. Chinese Guideline on the Primary Prevention of Cardiovascular Diseases. *Cardiology Discovery* **1**(2021).
4. Pulit, S.L., *et al.* Meta-analysis of genome-wide association studies for body fat distribution in 694 649 individuals of European ancestry. *Hum Mol Genet*

- 28, 166–174 (2019).
5. Brandts, J., *et al.* Optimal implementation of the 2019 ESC/EAS dyslipidaemia guidelines in patients with and without atherosclerotic cardiovascular disease across Europe: a simulation based on the DA VINCI study. *Lancet Reg Health Eur* **31**, 100665 (2023).
 6. Cholesterol Treatment Trialists, C., *et al.* The effects of lowering LDL cholesterol with statin therapy in people at low risk of vascular disease: meta-analysis of individual data from 27 randomised trials. *Lancet* **380**, 581–590 (2012).
 7. Zhao, D., Liu, J., Xie, W. & Qi, Y. Cardiovascular risk assessment: a global perspective. *Nat Rev Cardiol* **12**, 301–311 (2015).
 8. Li, X., *et al.* Cardiovascular risk factors in China: a nationwide population-based cohort study. *Lancet Public Health* **5**, e672–e681 (2020).
 9. Li, J.J., *et al.* 2023 Chinese guideline for lipid management. *Front Pharmacol* **14**, 1190934 (2023).
 10. Shang, H.C., W. X.; Pang, P. S.; Zhang, J.; Wang, L. L.; He, X. K.; Huang, X. Z. Reference intervals for common tests of liver function, electrolytes and blood cell analysis of Chinese adults (In Chinese). *Chin J Lab Med* **36**, 393–394 (2013).
 11. Chinese National Health, C. Reference intervals for common clinical biochemistry tests. *National Health Committee of the People’s Republic of China* (2012).
 12. Hernán, M.A., McAdams, M., McGrath, N., Lanoy, E. & Costagliola, D. Observation plans in longitudinal studies with time-varying treatments. *Statistical Methods in Medical Research* **18**, 27–52 (2009).
 13. Díaz, I., Williams, N., Hoffman, K.L. & Schenck, E.J. Nonparametric Causal Effects Based on Longitudinal Modified Treatment Policies. *J Am Stat Assoc* **118**, 846–857 (2023).
 14. Díaz, I., Hoffman, K.L. & Hejazi, N.S. Causal survival analysis under competing risks using longitudinal modified treatment policies. *Lifetime Data Anal* **30**, 213–236 (2024).
 15. Hoffman, K.L., *et al.* Comparison of a Target Trial Emulation Framework vs Cox Regression to Estimate the Association of Corticosteroids With COVID-19 Mortality. *JAMA network open* **5**, e2234425 (2022).
 16. Chou, R., *et al.* Statin Use for the Primary Prevention of Cardiovascular Disease in Adults: Updated Evidence Report and Systematic Review for the US Preventive Services Task Force. *Jama* **328**, 754–771 (2022).
 17. Hernán, M.A. & Robins, J.M. Per-Protocol Analyses of Pragmatic Trials. *N Engl J Med* **377**, 1391–1398 (2017).
 18. Hernan, M.A. Methods of public health research – Strengthening causal inference from observational data. *N Engl J Med* **385**, 1345–1348 (2021).
 19. Wang, M., Liu, J. & Zhao, D. New risk assessment tool of atherosclerotic cardiovascular disease for Chinese adults. *Zhonghua Xin Xue Guan Bing Za Zhi* **46**, 87–91 (2018).

20. Liu, J., *et al.* Predictive value for the Chinese population of the Framingham CHD risk assessment tool compared with the Chinese Multi-Provincial Cohort Study. *JAMA* **291**, 2591-2599 (2004).
21. Qi, Y., *et al.* Long-Term Cardiovascular Risk Associated With Stage 1 Hypertension Defined by the 2017 ACC/AHA Hypertension Guideline. *J Am Coll Cardiol* **72**, 1201-1210 (2018).
22. Wu, Z., *et al.* Sino-MONICA project: a collaborative study on trends and determinants in cardiovascular diseases in China, Part i: morbidity and mortality monitoring. *Circulation* **103**, 462-468 (2001).
23. Hernán, M.A. & Robins, J.M. Using Big Data to Emulate a Target Trial When a Randomized Trial Is Not Available. *Am J Epidemiol* **183**, 758-764 (2016).
24. Zafeiropoulos, S., *et al.* Reinforcing adherence to lipid-lowering therapy after an acute coronary syndrome: A pragmatic randomized controlled trial. *Atherosclerosis* **323**, 37-43 (2021).
25. Chinese Society of Cardiology of Chinese Medical Association, C.D.P.a.R.C.o.C.A.o.R.M., Cardiovascular Disease Committee of Chinese Association of Gerontology and Geriatrics, *et al.* Chinese Society of Cardiology of Chinese Medical Association, Cardiovascular Disease Prevention and Rehabilitation Committee of Chinese Association of Rehabilitation Medicine, Cardiovascular Disease Committee of Chinese Association of Gerontology Geriatrics, Thrombosis Prevention Treatment Committee of Chinese Medical Doctor Association. Chinese guideline on the primary prevention of cardiovascular diseases. *Zhonghua Xin Xue Guan Bing Za Zhi* **48**, 1000-1038 (2020).
26. Young, J.G., Stensrud, M.J., Tchetgen Tchetgen, E.J. & Hernan, M.A. A causal framework for classical statistical estimands in failure-time settings with competing events. *Stat Med* **39**, 1199-1236 (2020).
27. Packard, C.J. & Ford, I. Long-term follow-up of lipid-lowering trials. *Current opinion in lipidology* **26**, 572-579 (2015).
28. Nayak, A., *et al.* Legacy effects of statins on cardiovascular and all-cause mortality: a meta-analysis. *BMJ open* **8**, e020584 (2018).
29. Bosch, J., *et al.* Lowering cholesterol, blood pressure, or both to prevent cardiovascular events: results of 8.7 years of follow-up of Heart Outcomes Evaluation Prevention (HOPE)-3 study participants. *European heart journal* **42**, 2995-3007 (2021).
30. O'Donoghue, M.L., *et al.* Long-Term Evolocumab in Patients With Established Atherosclerotic Cardiovascular Disease. *Circulation* **146**, 1109-1119 (2022).
31. Schwartz, G.G., *et al.* Transiently achieved very low LDL-cholesterol levels by statin and alirocumab after acute coronary syndrome are associated with cardiovascular risk reduction: the ODYSSEY OUTCOMES trial. *European heart journal* **44**, 1408-1417 (2023).
32. Ford, I., Murray, H., McCowan, C. & Packard, C.J. Long-Term Safety and Efficacy of Lowering Low-Density Lipoprotein Cholesterol With Statin Therapy: 20-Year Follow-Up of West of Scotland Coronary Prevention Study. *Circulation*

- 133, 1073–1080 (2016).
33. Genest, J. Transient very low LDL-C levels: a legacy effect for cardiovascular prevention? *European heart journal* **44**, 1418–1420 (2023).
 34. Hernán, M.A., Hernández-Díaz, S. & Robins, J.M. Randomized trials analyzed as observational studies. *Annals of internal medicine* **159**, 560–562 (2013).
 35. Toh, S. & Hernán, M.A. Causal inference from longitudinal studies with baseline randomization. *The international journal of biostatistics* **4**, Article 22 (2008).
 36. Baigent, C., *et al.* Efficacy and safety of cholesterol-lowering treatment: prospective meta-analysis of data from 90,056 participants in 14 randomised trials of statins. *Lancet* **366**, 1267–1278 (2005).
 37. Hernán, M.A. The hazards of hazard ratios. *Epidemiology (Cambridge, Mass.)* **21**, 13–15 (2010).
 38. Uno, H., *et al.* Moving beyond the hazard ratio in quantifying the between-group difference in survival analysis. *J Clin Oncol* **32**, 2380–2385 (2014).
 39. Yang, Z. & Yin, G. An alternative approach for estimating the number needed to treat for survival endpoints. *PLoS One* **14**, e0223301 (2019).
 40. McGrath, S., *et al.* gfoRmula: An R Package for Estimating the Effects of Sustained Treatment Strategies via the Parametric g-formula. *Patterns (N Y)* **1**(2020).
 41. Mihaylova, B., *et al.* The effects of lowering LDL cholesterol with statin therapy in people at low risk of vascular disease: meta-analysis of individual data from 27 randomised trials. *Lancet (London, England)* **380**, 581–590 (2012).
 42. García-Albéniz, X., Hsu, J. & Hernán, M.A. The value of explicitly emulating a target trial when using real world evidence: an application to colorectal cancer screening. *Eur J Epidemiol* **32**, 495–500 (2017).

Point-by-Point Responses to Comments from Reviewers on Manuscript

NCOMMS-24-03531A

Thank you very much for the in-principle acceptance of our manuscript. Considering all your comments, we have revised our manuscript, which led to a better exposition of our work. Our point-by-point responses to your comments are given below, with your original comments copied in *Italics* for your convenience. Any material from the revised manuscript is in **bold**, surrounded by quotation marks.

REVIEWERS' COMMENTS

Comments from Reviewer #1:

1. *I think the paper is substantially improved and is much clearer now.*

Response: Thank you for your very positive comments.

2. *I would like to suggest that the authors add a bit more detail on the method of imputing missing data across time. As is, it is not clear if data were simply carried forward from the last observation or if a (multiple) imputation approach was used.*

Response: Thank you for your comments. We detailed the imputation methods for those missing data over the study period on page 17, lines 469-474, as follows: “**When the time-varying variables were not measured, we imputed missing values using the most recently measured values of the in-person follow-up examination since the last measurement over the study period (i.e., the last observation carried forward), as recommended by Hernán, et al.** ^[54]”

Reference:

^[54] Hernán, M.A., McAdams, M., McGrath, N., Lanoy, E. & Costagliola, D. Observation plans in longitudinal studies with time-varying treatments. *Statistical Methods in Medical Research* **18**, 27-52 (2009).

Comments from Reviewer #2:

3. *The authors have addressed all my comments and suggestions, the current manuscript reads clearly. I have a few minor comments:*

The current title of the manuscript is set up as a conclusion. To maintain objectivity and encourage critical engagement of readers, consider revising the title to a more neutral and descriptive format that reflects the study's aim/design rather than interpreting the results. Otherwise, it can lead to misinterpretations, considering that the interpretation requires understanding the limitations and potential sources of bias in the analysis.

Responses: Thank you for your invaluable comments. We have revised the title of our manuscript from “Treat-to-target cholesterol-lowering interventions reduce cardiovascular disease and all-cause mortality risk in the community-dwelling population: a target trial emulation using data from a 29-year cohort study” to “**Effectiveness of treat-to-target cholesterol-lowering interventions on cardiovascular disease and all-cause mortality risk in the community-dwelling population: a target trial emulation using data from a 29-year cohort study**”.

5. *Page 4, lines 86 – 91. The sentence “Lastly, randomized controlled trials, particularly sequentially randomized experiments randomly assigned treat-to-target cholesterol lowering” is a bit lengthy but doesn’t convey as much information as it could. I think it would make a stronger point to say that we mostly care about per protocol effects, rather than the ITT (probably what most RCT’s report), but for that we need to account for time-varying confounding feedback. While the idea of sequential randomized experiments helps convey the need to remove the confounder feedback loops, it does not necessarily mean that the interest is on long-term effects.*

Responses: Thank you for your important comments. We have highlighted the point of per-protocol effects and the time-varying confounding feedback issues when estimating treat-to-target effects by revising sentences on page 4, lines 75-83 from “Lastly, randomized controlled trials, particularly sequentially randomized

experiments randomly assigned treat-to-target cholesterol-lowering intervention at each visit during follow-up according to previous risk profiles and intervention effects to each participant, are required for quantifying the long-term effects, which are typically not feasible for practical and ethical reasons.” to “**Lastly, treat-to-target cholesterol-lowering interventions involve previous risk profiles and intervention effects at each follow-up visit, conferring per-protocol effects. Thus, the estimation of treat-to-target effects requires accounting for time-varying confounding feedback. Sequentially randomized controlled experiments randomly assigned treat-to-target cholesterol-lowering interventions at each visit during follow-up according to previous risk profiles and intervention effects can appropriately address time-varying confounding by adjusting for a set of minimal sufficient confounders, but this protocol is typically not feasible for practical and ethical reasons** ^[29].”

Reference:

[29] Hernán MA, Robins JM (2020). Causal Inference: What If. Boca Raton: Chapman & Hall/CRC. Chapter 19, p258-269.

6. *Line 297, author’s start a sentence with “Finally” that is followed by another sentence with “Fourth”.*

Responses: Thank you for your comments. We have updated the sentences accordingly.

7. *In the limitations section, the authors need to include that the intervals between waves are very long, and although the g-formula is set to “intervene” on a yearly basis, these are effectively happening whenever data is updated (every 5 – 10 years). I think this is hinted in line 305 where they mention more regular repeated measurements, but they should be more explicit of the limitation.*

Responses: Thank you for your invaluable comments. We have explained this point by revising sentences on page 11, lines 291-295 from “Finally, our findings might have been underpowered and imprecise due to the relatively small sample size. A

large-scale longitudinal study with more regular repeated measurements of traditional cardiovascular risk profiles, lifestyle, medication, and disease history (e.g., major chronic diseases) would further triangulate our results.” to “**Finally, our findings might have been underpowered and imprecise due to the relatively small sample size and large intervals between follow-up visits over the study period. A large-scale longitudinal study with more frequent measurements of traditional cardiovascular risk profiles, lifestyle, medication, and disease history (e.g., major chronic disease) would further triangulate our results.**”

8. *Furthermore, participants who were not from Beijing and attended wave 0 and 2 would have a 15-year gap between measurements. Authors must explain why this decision is more reasonable than restricting analysis to those participants from Beijing area only, and what assumptions are they willing to make with this decision (in terms of carrying forward data for such a long time).*

Responses: Thank you for your comments. We included participants beyond the Beijing area mainly because only two-fifths of study participants were recruited from the Beijing area. Restricting participants from the Beijing area could induce selection bias and tend to bias the target-to-target effects toward the null. Moreover, including participants beyond the Beijing area increased the sample size and study power to obtain a more precise target-to-target estimate, considering the relatively large intervals between follow-up visits ^[48].

Reference:

^[48] García-Albéniz, X., Hsu, J. & Hernán, M.A. The value of explicitly emulating a target trial when using real world evidence: an application to colorectal cancer screening. *Eur J Epidemiol* **32**, 495-500 (2017).

Comments from Reviewer #3:

9. *The authors are thanked for their thoughtful responses. This reviewer remains concerned that the same limitations that were raised are what they are. There are fundamental limitations to simulation, they don't overcome the limitations of real world clinical trials and the other reviewers have highlighted other factors that are real - simulation assumes consistency, non confounding and perfection - this clinical trialist can humbly aspire to work in such a world.*

Response: Thank you for your comments. We fully agree with you that target trial emulation or simulation involves assumptions. Following suggestions from you and other reviewers, we have carefully discussed the potential impacts on our results when violating assumptions on consistency, non-confounding, and perfection in the limitation section on pages 10-11.

10. *The authors should tone down the suggestion that all approaches to LDL-C lowering, including diet, will produce such findings. The reality is that dietary effects on LDL-C are modest and it is misleading of the authors to suggest that their simulated findings would be achieved by people simply adopting a diet.*

Responses: Thank you for your important comments. We have revised the sentence “However, we didn’t explain how cholesterol levels would be lowered, which means the observed protective estimates can be results from either lifestyle or pharmaceutical interventions or any other approaches, making the consistency assumption dubious in the current study” on pages 10-11, lines 279-282 as below “**However, we didn’t explain how cholesterol levels would be lowered, which means the observed protective estimates are overall interventional effects, making the consistency assumption dubious in the current study.**”